# Examination of the parameters controlling the triple oxygen isotope composition of grass leaf water and phytoliths at a Mediterranean site: A model-data approach

Claudia Voigt[1,2], Anne Alexandre[1], Ilja M. Reiter[3], Jean-Philippe Orts[4], Christine Vallet-Coulomb[1], Clément Piel[5], Jean-Charles Mazur[1], Julie C. Aleman[1], Corinne Sonzogni[1], Helene Miche[1], Jérôme Ogée[6]

[1]Aix Marseille Université, CNRS, IRD, INRAE, CEREGE, 13545 Aix-en-Provence, France
[2]Present address: University of Almería, Department of Biology and Geology, 04120 Cañada de San Urbano, Almería, Spain
[3]Research Federation ECCOREV, FR3098, CNRS, 13545 Aix-en-Provence, France
[4]IMBE, CNRS, Université d'Avignon, Aix-Marseille Université, IRD, 13397 Marseille, France
[5]ECOTRON Européen de Montpellier, UAR 3248, CNRS, Campus de Baillarguet, 34980 Montferrier-sur-Lez, France
[6]INRAE, Bordeaux Sciences Agro, UMR ISPA, 33140 Villenave d'Ornon, France

*Correspondence to*: Claudia Voigt (cvoigt@ual.es), Anne Alexandre (alexandre@cerege.fr)

## Abstract

Triple oxygen isotopes ($^{17}$O-excess) of water are useful to trace evaporation at the soil-plant-atmosphere interface. The $^{17}$O-excess of plant silica, i.e. phytoliths, inherited from leaf water, was previously calibrated in growth chambers as a proxy of atmospheric relative humidity (RH). Here, using a model-data approach, we examine the parameters that control the triple oxygen isotope composition of bulk grass leaf water and phytoliths *in natura*, at the O₃HP experimental platform located in the French Mediterranean area. A grass plot was equipped to measure for one year, all environmental and plant physiological parameters relevant for modelling the isotope composition of the grass leaf water. In particular, the triple oxygen and hydrogen isotope composition of atmospheric water vapor above the grass was measured continuously using a cavity ring-down spectrometer, and the grass leaf temperature was monitored at plot-scale using an infra-red (IR) radiometer. Grass leaves were collected in different seasons of the year and over a 24-hour period in June. Grass leaf water was extracted by cryogenic vacuum distillation and analyzed by isotope ratio mass spectrometry (IRMS). Phytoliths were analyzed by IR-laser fluorination-IRMS after chemical extraction. We showed that the traditional Craig-Gordon steady state model modified for grass leaves reliably predicts the triple oxygen isotope composition of leaf water during daytime but is sensitive to uncertainties on the leaf-to-air temperature difference. Deviations from isotope steady state at night are well represented in the triple oxygen isotope system and predictable by a non-steady state model. The $^{17}$O-excess of phytoliths confirms the applicability of the $^{17}$O-excess$_{phyto}$ vs RH equation established in previous growth chamber experiments. Further, it recorded average daytime RH over the growth period rather than daily RH, related to low transpiration and silicification during the night. This model-data approach highlights the utility of the triple oxygen isotope system to improve the understanding of water exchange at the soil-plant-atmosphere interface. The *in natura* experiment underlines the applicability of $^{17}$O-excess of phytoliths as a RH proxy.

# 1 Introduction

Continental atmospheric relative humidity (RH) is a key factor of soil evaporation, transpiration, dryness stress and ecosystem productivity (Grossiord et al., 2020; Liu et al., 2021; López et al., 2021). However, RH is estimated with low precision in the Earth system models (IPCC, 2013; Tierney et al., 2020). Long-term data beyond the instrumental period is needed to improve the representation of RH in these models. Leaf organic and mineral compounds formed during plant growth, such as cellulose, n-alkanes of leaf waxes, or phytoliths are used as past climate indicators when preserved in soils, sediments or peat (Helliker and Ehleringer, 2002a, b; Kahmen et al., 2011, 2013; Zech et al., 2014; Tuthorn et al., 2015; Alexandre et al., 2018, 2019; Outrequin et al., 2021; Garcin et al., 2012, 2022). To accurately interpret the isotope signal of these compounds in terms of paleoclimate, their relationship to that of leaf water and the factors driving their isotope variability need to be determined. Regarding the phytolith isotope signature, previous calibrations have often been performed in controlled environmental conditions, not representative of the diurnal, daily and seasonal climate variations encountered in the natural environment (Alexandre et al., 2018, 2019; Outrequin et al., 2021). Therefore, the question of the time span (seasonal vs annual, diurnal vs daily) integrated in the phytolith isotope composition remains open.

Leaf waters generally show higher $\delta^2H$, $\delta^{18}O$, and lower d-excess [$= \delta^2H - 8\,\delta^{18}O$] than meteoric waters due to significant evaporative fractionation during transpiration. The magnitude of this isotope fractionation can be predicted by the isotope-evaporation model developed by Craig and Gordon (1965), and later adapted to leaf transpiration (Dongmann et al., 1974; Farquhar and Cernusak, 2005). This model (hereafter referred to as the C-G model) considers three main processes occurring in the boundary layer of the leaf during transpiration: (i) liquid water-water vapor equilibrium at the boundary layer interface, (ii) diffusion of water vapor from the evaporative sites in the leaf to the surrounding air, and (iii) back-diffusion of atmospheric water vapor to the leaf (Craig and Gordon, 1965; Farquhar et al., 2007; Cernusak et al., 2016). The C-G model is based on the steady state assumption, i.e. all water that is lost by evaporation is continuously replenished by xylem water. This assumption neglects small diurnal changes in leaf water content that are expected to result in only 3 % error in the predicted leaf water $\delta^{18}O$ enrichment (Farris and Strain, 1978; Farquhar and Cernusak, 2005). The C-G model also assumes isotope steady state, so that the isotope composition of transpired water matches that of source (xylem) water. To take into account the advection of less evaporated stem water to the evaporation site, as well as the diffusion of the evaporating water back to the leaf lamina, a transpiration-dependent correction, called the Péclet effect, can be added to the C-G model (e.g., Buhay et al., 1996; Helliker and Ehleringer, 2000; Roden et al., 2000; Farquhar and Gan, 2003; Farquhar and Cernusak, 2005; Ripullone et al., 2008; Treydte et al., 2014). For grasses, a two-pool model, including a pristine water pool that coincides to the xylem tissues and an evaporated water pool that corresponds to leaf lamina water has been found to best represent bulk leaf water (Liu et al., 2017; Hirl et al., 2019; Barbour et al., 2021). This mixing effect is independent from transpiration, so that a two-endmember mixing equation is combined with the C-G model (Leaney et al., 1985).

Although the modelling approaches described above reproduce the observed trends in the isotope composition of bulk leaf water, discrepancies between modeled and observed $\delta^{18}O$ values as high as 6 ‰ have been reported (e.g., Flanagan et al., 1991; Gan et al., 2002; Loucos et al., 2014; Song et al., 2015; Cernusak et al., 2016; Bögelein et al., 2017). These discrepancies can arise from uncertainties in key parameters of the C-G model that are difficult to measure, such as the isotope composition of atmospheric water vapor and the difference between leaf temperature and air temperature (Cernusak et al., 2002; Flanagan and Farquhar, 2014; Li et al., 2017; Alexandre et al., 2018). The isotope composition of atmospheric water vapor varies greatly in space and time, in principle depending on the climate conditions in the air mass source region and processes affecting the air mass during transport, including rainout, moisture recycling, and mixing. In the absence of direct measurements, the isotope composition of atmospheric water vapor is often estimated, assuming isotope equilibrium with local precipitation. This assumption can be valid on monthly timescales, but large deviations can occur on daily or hourly timescales (Jacob and Sonntag, 1991; Lee et al., 2006; Aemisegger et al., 2015; Graf et al., 2019; Penchenat et al., 2020). Variations in leaf temperature slightly influence the equilibrium isotope fractionation at the liquid-vapor interface. More importantly the deviation of the leaf temperature from the air temperature ($\Delta T_{leaf-air}$) determines the water vapor pressure gradient between the leaf and the atmosphere, one of the major controls of the isotope composition of bulk leaf water. However, large spatial and temporal variability of leaf temperatures complicate measurement or accurate estimation of $\Delta T_{leaf-air}$. Ultimately, deviations from isotope steady state resulting from low stomatal conductance ($g_s$) and transpiration rate and thus long leaf water residence time in the mesophyll cells, notably occurring at night or during drought, can also account for model-data discrepancies (Cuntz et al., 2007; Ogée et al., 2007; Cernusak et al., 2016; Wang et al., 2018).

Recent analytical advances enable the analysis of $\delta^{17}O$ in addition to $\delta^{18}O$, allowing to derive the secondary parameter $^{17}O$-excess [= $\delta'^{17}O - 0.528\ \delta'^{18}O$ with $\delta' = \ln(\delta+1)$], with 0.528 being the slope of the Global Meteoric Water Line (GMWL) (Luz and Barkan, 2010). The small variations in $^{17}O$-excess are usually reported in 'per meg', i.e. 0.001 ‰. As d-excess, $^{17}O$-excess of liquid water decreases with increasing evaporation. However, in contrast to $\delta^{18}O$, $\delta^2H$ or d-excess, the $^{17}O$-excess is weakly affected by temperature changes and Rayleigh distillation. This is due to its low sensitivity to equilibrium isotope fractionation between liquid water and water vapor (Barkan and Luz, 2005). Consequently, $^{17}O$-excess varies little in meteoric water, which feeds the soil water taken up by the plants and is also assumed to vary little in atmospheric water vapor (Luz and Barkan, 2010; Aron et al., 2021; Surma et al., 2021). The $^{17}O$-excess of bulk leaf water is thus essentially controlled by the molecular diffusion of water vapor between the leaf and the atmosphere during transpiration (Barkan and Luz, 2007). The extent of this process depends mainly on the water pressure gradient between the leaf and the atmosphere. The few existing studies on $^{17}O$-excess of bulk leaf water showed that it is inversely related to RH. Discrepancies between modeled and observed $^{17}O$-excess values higher than 100 per meg have been reported (Li et al., 2017; Alexandre et al., 2018; Outrequin et al., 2021). These discrepancies have been attributed to deviations from isotope steady state in the early morning hours (Li et al., 2017) and uncertainty in the estimates of leaf temperature and the isotope composition of atmospheric water vapor (Li et al., 2017;

Alexandre et al., 2018). Large discrepancies observed by Li et al. (2017) may also result from neglecting potential mixing of evaporated and non-evaporative grass leaf water pools.

Phytoliths are micrometric silica particles that form in temperature-dependent isotope equilibrium with water in living plant tissues within a few hours to days (Perry et al., 1987). In grasses, the majority of phytoliths forms in sheaths and leaves, due to concentration of solutes by transpiration (e.g., Webb and Longstaffe, 2000, 2002). Phytolith morphological assemblages recovered from soils and sediments are used to reconstruct vegetation changes and qualitatively inform on climatic conditions at the time of soil formation (Bremond et al., 2005; Aleman et al., 2012; Nogué et al., 2017). Previous studies investigated the potential of $\delta^{18}O$ of phytoliths as a proxy for past temperature (Webb and Longstaffe, 2000, 2002, 2006; Alexandre et al., 2012). However, accurate temperature reconstruction using this proxy requires an independent estimate of the $\delta^{18}O$ of soil water, and an estimate of the effect of RH and transpiration on $\delta^{18}O$ of leaf water. These studies have also shown the dependency of $\delta^{18}O$ of phytoliths on RH, but its utility to reconstruct past RH has not been further explored given the large number of factors influencing $\delta^{18}O$ of precipitation, soil, and leaf water. Recent studies in growth chambers and at natural sites demonstrated that unlike the $\delta^{18}O$, the $^{17}O$-excess of phytoliths ($^{17}O$-excess$_{phyto}$), inherited from the $^{17}O$-excess of leaf water, is primarily controlled by RH around the plant, according to a gradient of $4.3 \pm 0.3$ per meg %$^{-1}$ (Outrequin et al., 2021). This relationship is independent of grass leaf length and vegetation type (Alexandre et al., 2018, 2019; Outrequin et al., 2021). Further, the $^{17}O$-excess$_{phyto}$ is not affected by changes in air temperature or atmospheric $CO_2$ levels (Outrequin et al., 2021).

In this study, using a model-data approach, we examined the parameters controlling the triple oxygen isotope composition of bulk grass leaf water and phytoliths at a natural site. For that purpose, a grass plot was equipped to measure for the course of one year, all environmental and plant physiological parameters relevant for modelling the isotope composition of the grass leaf water. In particular, the triple oxygen and hydrogen isotope composition of atmospheric water vapor above the grass was measured continuously over the year using a cavity ring-down spectrometer (CRDS), and the grass leaf temperature was monitored at plot-scale using an infra-red (IR) radiometer. Grass leaf blades were collected at midday on eight days in different seasons of the year and over a 24-hour period in June for triple oxygen and hydrogen isotope analysis of bulk leaf waters. In addition, grass leaf blades were harvested in spring, summer and autumn for phytolith extraction and triple oxygen isotope analysis to examine which RH average is recorded in $^{17}O$-excess$_{phyto}$ of phytolith assemblages that are formed over growth periods of several months.

## 2 Materials & Methods

### 2.1 Experimental setup

The AnaEE *in natura* experimental platform O₃HP is located about 100 km north of Marseille (France) at an altitude of 680 m above sea level (43.935° N, 5.711° E). On 14 February 2021, seeds of the C3 grass species *Festuca arundinacea*, also referred to as tall fescue, were sown (8 g m$^{-2}$) on a 5.5 m$^2$ plot in the understory of an oak-dominated forest. The same grass species

was used for the calibration of the relationship between [17]O-excess$_{phyto}$ and RH in growth chamber experiments (Alexandre et al., 2018, 2019; Outrequin et al., 2021). Potting soil was added to the shallow calcaric leptosol (IUSS Working Group WRB, 2015; Belviso et al., 2016) and supplied with ~ 50 g m$^{-2}$ organic fertilizer (Engrais Gazon, Neudorff, Emmerthal, Germany) and 2.7 g m$^{-2}$ SiO$_2$ (General Hydroponics Mineral Magic, Terra Aquatica, Fleurance, France) to ensure a sufficient amount of nutriments and bio-available silica.

The experimental plot was automatically irrigated with tap water (30 mm d$^{-1}$) from 04 March 2021 until the end of the experiment on 23 November 2021 to avoid water stress in the grasses. The potential evaporation from the grass plot (2–4 mm d$^{-1}$) estimated using the Penman-Monteith equation (Monteith, 1965) was an order of magnitude lower than the irrigation rate. Therefore, we assume that soil water evaporation was negligible and had no impact on the isotope composition of leaf water. An aliquot of the irrigation water was collected in an evaporation-free water collector (Rain Sampler 1, Palmex d.o.o., Zagreb, Croatia; Gröning et al., 2012), that was sampled weekly. Precipitation was collected on an event-based interval using a second water collector of the same type. Both collectors were emptied and dried after sampling. For isotope analysis of atmospheric water vapor, the air at 0.4 m above the grass plot was pumped continuously (N 86 KN.18, KNF DAC GmbH, Hamburg, Germany) to a Picarro L2140-i CRDS (Picarro Inc., California, USA), installed in an air-conditioned cabin on the experimental site. The air was passed through a 11.5 m long and 1/4 " wide PFA tube (PFA-T4-062-100, Swagelok, Ohio, USA), at a flow rate of 5 L min$^{-1}$. The tubing was insulated and heated to prevent condensation of the water vapor. A funnel covered by a net was placed at the inlet for protection from rain and suction of insects and large aerosol particles.

The following climate parameters were measured on the experimental site: Global solar radiation at 6 m above ground (LI-200, LI-COR Biosciences Inc., Nebraska, USA), precipitation amount (15189 H, LAMBRECHT meteo GmbH, Göttingen, Germany), RH and atmospheric temperature (T$_{air}$) at 60 cm height next to the grass plot (HMP155, Vaisala Oyj, Vantaa, Finland), atmospheric temperature at 5 cm above the ground (T$_{ground}$) (DTS12, Vaisala Oyj, Vantaa, Finland), soil water content and soil temperature at ~ 5 cm depth (CS655, Campbell Scientific Inc, Logan, Utah, USA), plot-scale grass leaf temperature (T$_{plot}$) (IR radiometer SI-411-SS, Apogee Instruments Inc., Utah, USA), and sky temperature (T$_{sky}$). T$_{plot}$ is the temperature integrated over the field of view of the IR radiometer that covered ~ 90 % of the grass plot surface. Each parameter was extracted in hourly resolution from the COOPERATE database (COOPERATE database, 2022).

On sampling days (Table 1), stomatal conductance (g$_s$) and transpiration, were monitored continuously over the day on a single grass leaf of 4–5 mm width using a Li-6400 XT gas exchange system (LI-COR Biosciences Inc., Nebraska, USA). To assess the spatial variability of g$_s$, this parameter was additionally measured hourly on the adaxial side of ten leaves of at least 3 mm width, randomly selected on the plot, using an AP4 porometer (Table S1; Delta-T Devices LTD, Cambridge, UK). In addition, leaf temperature (T$_{leaf}$) was measured in situ on the adaxial side of ten grass leaves, randomly selected, in one-hour intervals using an Optris CT IR thermometer (Table S2; Optris GmbH, Berlin, Germany). T$_{plot}$ and T$_{leaf}$ measurements were corrected for emissivity of the grass canopy, considering the tree canopy gap fraction:

$$T_{plot} \text{ or } T_{leaf} = \sqrt[4]{\frac{T_{raw}^4 - (1-\varepsilon)\cdot(\alpha\cdot T_{sky} + (1-\alpha)\cdot T_{canopy})^4}{\varepsilon}} \tag{1}$$

where $\varepsilon$ is the emissivity of the grass canopy ($\varepsilon = 0.95$; Apogee Instruments Inc, 2022) and $\alpha$ is the tree canopy gap fraction, which is estimated to be 0.3 throughout the experimental period. $T_{raw}$ is the temperature recorded by the sensor, $T_{sky}$ is the sky temperature and $T_{canopy}$ is the canopy temperature, which is assumed to equal to $T_{air}$.

## 2.2 Sampling

Leaf blades of *F. arundinacea* were collected at midday on eight days in May, July, August, October, and November 2021 (Table 1), as well as every ~ 1.5 h over a 24-hour period from 14–15 June 2021. About ten fully developed, not senescent leaf blades from different tillers evenly distributed over the grass plot were immediately transferred to 12 mL Exetainer vials (Labco, High Wycombe, UK), and stored in a fridge until water extraction and isotope analysis.

Three grass regrowths were monitored in spring (17 February–20 May 2021), summer (15 June–27 August 2021), and autumn (27 August–23 November 2021) (Table 2). Each regrowth started after the grasses had been cut above the sheath at 2–4 cm height. Grass heights were measured at monthly intervals. At the end of each regrowth, the grass leave blades from the entire plot were harvested and dried at 50 ºC. Between 120 and 150 g of dry matter were obtained for phytolith extraction and analysis.

**Table 1:** *F. arundinacea* leaf water isotope composition ($\delta^{18}O$, $^{17}O$-excess, and d-excess), stomatal conductance ($g_s$) and transpiration (E) measured on a single leaf blade using the LI-COR gas exchange system, atmospheric temperature ($T_{air}$) and relative humidity (RH) at 60 cm height next to the grass plot, plot-scale grass leaf temperature ($T_{plot}$), and the ratio of atmospheric vapor pressure at 60 cm height and saturation vapor pressure at $T_{plot}$ (h), averaged over 30 minutes before sampling on 8 days at midday between May and November 2021 and 14 samplings during a 24-hour period from 14–15 June 2021. The sample ID indicates 'sampling location_plant_species_sample type_sampling date_sampling time'. Plant species 'FA' denotes the C3 grass *Festuca arundinacea*, sampling date is in the format YYYYMMDD and sampling time in UTC. $\Delta T_{leaf\text{-}air} = T_{plot} - T_{air}$.

| Sample ID | E (mmol $m^{-2}$ $s^{-1}$) | $g_s$ (m mol $m^{-2}$ $s^{-1}$) | $T_{air}$ (°C) | $T_{plot}$ (°C) | $\Delta T_{leaf\text{-}air}$ (°C) | RH (%) | h (%) | $\delta^{18}O$ (‰) | $^{17}O$-excess (per meg) | d-excess (‰) |
|---|---|---|---|---|---|---|---|---|---|---|
| **Midday samples** | | | | | | | | | | |
| O3HP_FA_leaf_20210503_1130 | 2.5 | 97 | 17.4 | 17.5 | 0.1 | 42 | 42 | 9.87 | -122 | -88.5 |
| O3HP_FA_leaf_20210520_1130 | 2.2 | 114 | 20.2 | 18.5 | -1.7 | 36 | 40 | 20.09 | -165 | -142.8 |
| O3HP_FA_leaf_20210722_1155 | 3.7 | 84 | 32.6 | 29.9 | -2.7 | 27 | 32 | 12.53 | -156 | -99.3 |
| O3HP_FA_leaf_20210826_1140 | 1.3 | 49 | 26.7 | 23.6 | -3.1 | 42 | 50 | 4.47 | -77 | -52.3 |
| O3HP_FA_leaf_20210827_1130 | – | – | 24.8 | 22.5 | -2.3 | 38 | 43 | 6.37 | -103 | -59.3 |
| O3HP_FA_leaf_20211022_1130 | – | – | 17.2 | 15.2 | -2.0 | 65 | 74 | 3.19 | -52 | -36.3 |
| O3HP_FA_leaf1_20211027_1130 | 1.1 | 107 | 16.2 | 14.6 | -1.6 | 64 | 71 | 2.80 | -43 | -34.4 |
| O3HP_FA_leaf_20211123_1230 | 1.4 | 127 | 13.9 | 11.7 | -2.2 | 62 | 71 | -0.05 | 17 | -31.0 |
| O3HP_FA_leaf2_20211123_1230 | 1.4 | 127 | 13.9 | 11.7 | -2.2 | 62 | 71 | 1.63 | -3 | -42.6 |
| **24-hour period** | | | | | | | | | | |
| O3HP_FA_leaf_20210614_1720 | – | – | 30.1 | 26.4 | -3.7 | 38 | 47 | 7.96 | -108 | -72.1 |
| O3HP_FA_leaf_20210614_1830 | 0.3 | 10 | 27.0 | 24.3 | -2.7 | 38 | 45 | 9.96 | -135 | -84.5 |
| O3HP_FA_leaf_20210614_1945 | 0.5 | 21 | 23.6 | 21.8 | -1.8 | 43 | 48 | 10.49 | -151 | -87.2 |
| O3HP_FA_leaf_20210614_2135 | 0.4 | 16 | 21.4 | 19.9 | -1.6 | 41 | 45 | 6.29 | -110 | -61.6 |
| O3HP_FA_leaf_20210615_0315 | 0.1 | 13 | 15.1 | 16.0 | 0.9 | 97 | 92 | 3.86 | -91 | -44.4 |
| O3HP_FA_leaf_20210615_0445 | 0.0 | 3 | 14.5 | 15.6 | 1.1 | 97 | 90 | 2.49 | -85 | -36.5 |
| O3HP_FA_leaf_20210615_0615 | 0.7 | 87 | 19.0 | 19.8 | 0.9 | 91 | 87 | 2.12 | -60 | -31.2 |
| O3HP_FA_leaf_20210615_0800 | 1.8 | 75 | 24.3 | 23.1 | -1.1 | 69 | 74 | 2.55 | -43 | -31.2 |
| O3HP_FA_leaf_20210615_0930 | 1.3 | 63 | 26.8 | 24.9 | -1.9 | 67 | 75 | 2.31 | -45 | -27.2 |
| O3HP_FA_leaf_20210615_1100 | 1.9 | 79 | 28.0 | 24.9 | -3.1 | 58 | 70 | 4.60 | -65 | -42.5 |
| O3HP_FA_leaf_20210615_1230 | 3.7 | 118 | 29.8 | 27.2 | -2.5 | 51 | 58 | 5.10 | -65 | -44.7 |
| O3HP_FA_leaf_20210615_1400 | 3.9 | 111 | 30.8 | 27.1 | -3.6 | 43 | 53 | 4.22 | -62 | -40.4 |
| O3HP_FA_leaf_20210615_1530 | 2.2 | 89 | 28.1 | 26.5 | -1.6 | 63 | 69 | 4.32 | -63 | -39.2 |
| O3HP_FA_leaf_20210615_1700 | 1.6 | 78 | 27.4 | 25.3 | -2.1 | 63 | 72 | 4.94 | -32 | -46.4 |

**Table 2:** Grass and phytolith descriptors, phytolith isotope composition, atmospheric temperature ($T_{air}$), plot-scale grass leaf temperature ($T_{plot}$), relative humidity (RH) and the ratio between actual atmospheric vapor pressure and saturation vapor pressure at $T_{plot}$ (h) for the three regrowth periods. Grass height = grass height at the harvest day, LC = proportion of long cell phytoliths on the amount of short and long cell phytoliths in the sample. The silification rate is inferred from the measured $SiO_2$ concentration in grass leaf blades harvested at the end of the regrowth and the length of the regrowth period, assuming a linear production rate (av. rate). Observed RH and h values are compared to estimated values using [17]O-excess$_{phyto}$ and Eqs. (6) and (7), respectively (RH$_{phyto}$ and h$_{phyto}$, respectively). SD = 1 standard deviation of four replicate measurements on two consecutive days. $\Delta T_{leaf-air} = T_{plot} - T_{air}$.

| Sample | spring | summer | autumn |
|---|---|---|---|
| Regrowth period | 17/02/2021–20/05/2021 | 15/06/2021–27/08/2021 | 27/08/2021–23/11/2021 |
| **Grass and phytolith descriptors** | | | |
| Grass height (cm) | 43 | 25 | 18 |
| Silification rate (% $SiO_2$ dry weight $d^{-1}$) | 2.7 | 5.2 | 5.9 |
| LC (%) | 30 | 46 | 70 |
| **Phytolith isotope composition** | | | |
| $\delta^{18}O_{phyto}$ (‰) | 36.6±0.2 | 35.9±0.5 | 34.3±0.6 |
| [17]O-excess$_{phyto}$ (per meg) | -256±2 | -263±4 | -234±3 |
| **Observed temperature and relative humidity parameters** | | | |
| $T_{air}$ daily (°C) | 9±3 | 22±2 | 13±4 |
| $T_{air}$ daytime (°C) | 12±3 | 24±3 | 16±4 |
| $T_{plot}$ daily (°C) | 9±3 | 21±2 | 13±4 |
| $T_{plot}$ daytime (°C) | 12±3 | 23±2 | 15±4 |
| $\Delta T_{leaf-air}$ daily (°C) | -0.1±1.0 | -0.6±0.6 | -0.1±0.5 |
| $\Delta T_{leaf-air}$ daytime (°C) | 0.3±1.2 | -1.1±0.8 | -0.7±0.5 |
| RH daily (%) | 71±15 | 64±10 | 81±10 |
| RH daytime (%) | 62±17 | 57±11 | 73±12 |
| h daily (%) | 71±14 | 66±8 | 81±10 |
| h daytime (%) | 61±17 | 61±9 | 76±11 |
| **Estimated RH and h** | | | |
| RH$_{phyto}$ (%) | 59 | 57 | 64 |
| h$_{phyto}$ (%) | 66 | 64 | 71 |
| **Difference between estimated and observed RH and h** | | | |
| RH$_{phyto}$-RH daily (%) | -12 | -6 | -17 |
| RH$_{phyto}$-RH daytime (%) | -4 | 0 | -9 |
| h$_{phyto}$-h daily (%) | -5 | -2 | -10 |
| h$_{phyto}$-h daytime (%) | 5 | 3 | -4 |

## 2.3 Extractions and isotope analyses

### 2.3.1 Irrigation water, precipitation, and atmospheric water vapor

A Picarro L2140-i CRDS (California, USA), operated in $^{17}$O Dual Liquid/Vapor mode was installed on-site for the experiment. The isotope composition and mixing ratio of water vapor in the air at 0.4 m above the grass plot was measured for 70 min every 140 min during the spring monitoring and every 280 min during the monitoring in summer and autumn. In between these measurements, the instrument was used for another experiment. The atmospheric water vapor data from the first 10 minutes of each measurement cycle were removed to account for memory effects and provide sufficient time to establish a stable baseline. The remaining 60 minutes were averaged. During the 24-hour monitoring, air sampling was performed continuously without interruption. Liquid water standard measurement runs were performed on a weekly basis. The mean of four measurement runs of liquid water standards was used to normalize the atmospheric water vapor isotope data to VSMOW-SLAP scale. The calibration protocol is described in detail by Voigt et al. (2022). In brief, three liquid water standards that covered the expected isotope range of atmospheric water vapor at the study site were analyzed at a water mixing ratio of 11000 ppmv using a Picarro autosampler system (A0325, Picarro Inc., California, USA) coupled to a high-precision vaporizer (A0211, Picarro Inc., California, USA). The liquid standards were injected in a dry air stream, produced by a lubricated mobile air compressor (MONTECARLO FC2, ABAC air compressors, Italy), further dried using two drierite columns combined with a dry ice trap (Voigt et al., 2022). Raw isotope compositions of the liquid standards of four consecutive measurement runs were averaged and then corrected to the water mixing ratio of the measured atmospheric water vapor, using the mean of three mixing ratio dependency functions that were determined on site for water mixing ratios between 3000 and 30000 ppmv in May 2021, October 2021 and January 2022 (Fig. A1). The precision of calibrated and integrated atmospheric water vapor data was determined using a Monte Carlo simulation (Voigt et al., 2022). Precision was better than $\pm$ 0.1 ‰, $\pm$ 0.2 ‰, $\pm$ 1.8 ‰ and $\pm$ 14 per meg, and $\pm$ 0.9 ‰ for $\delta^{17}$O, $\delta^{18}$O, $\delta^{2}$H, $^{17}$O-excess, and d-excess, respectively.

A second Picarro L2140-i CRDS operated in $^{17}$O-High Precision mode was used at CEREGE to analyze the isotope composition of irrigation water and precipitation. Isotope analyses, correction of memory effects and VSMOW-SLAP scaling were performed following Vallet-Coulomb et al. (2021). The external reproducibility of a quality control standard (1 standard deviation (SD), n = 12) measured along with the samples in each sequence was $\pm$ 0.02 ‰, $\pm$ 0.03 ‰, $\pm$ 0.3 ‰, $\pm$ 6 per meg, and $\pm$ 0.1 ‰ for $\delta^{17}$O, $\delta^{18}$O, $\delta^{2}$H, $^{17}$O-excess, and d-excess, respectively.

### 2.3.2 Grass leaf water

Grass leaf water was extracted by cryogenic vacuum distillation (static pressure < 10 Pa) with sample vials placed in the vacuum line and immersed in a heated water bath for 3 h with a final target temperature set to 80 °C (attained within 45 min of extraction). A detailed description of the system design is given by Barbeta et al. (2022). Water extraction yield was derived by comparing the volume of water collected (in mL) and the difference of sample weights before and after water extraction

(with the exetainer and converted in equivalent mL of water). For our sample set, the average water extraction yield was 103 ± 5 % (102 ± 3 % without one outlier) and average extracted volume was 0.5 ± 0.2 mL, with only one extraction volume below 0.3 mL. Thus, methodological uncertainties linked to cryogenic vacuum distillation should be negligible (Diao et al. 2022). Isotope analysis of grass leaf waters was performed at the University of Cologne. For triple oxygen isotope analysis, pure $O_2$ liberated from grass leaf waters by fluorination was introduced in a Thermo Fisher Scientific MAT 253 dual-inlet mass spectrometer (Massachusetts, USA), following the procedure described by Surma et al. (2015). The reproducibility (1 SD, n = 2) of $\delta^{17}O$, $\delta^{18}O$ and $^{17}O$-excess measurements was better than ± 0.15 ‰, ± 0.30 ‰ and ± 11 per meg, respectively. Hydrogen isotope ratios were determined by high-temperature carbon reduction in a pyrolysis elemental analyzer (HEKAtech GmbH, Wegberg, Germany), coupled to the mass spectrometer. The reproducibility (1 SD, n = 3) of $\delta^2H$ measurements was always better than 1.1 ‰. An intercomparison of water analysis at CEREGE and the University of Cologne was performed. The results are presented in Table S3. Differences between the laboratories were lower than 0.2 ‰, 0.3 ‰, 1.1 ‰, 14 per meg, and 1.6 ‰ for $\delta^{17}O$, $\delta^{18}O$, $\delta^2H$, $^{17}O$-excess, and d-excess, respectively. Similar differences were found in an intercomparison between the two Picarro CRDS instruments (Alexandre et al., 2018).

### 2.3.3 Phytoliths

The silica contents of harvested grass leaf blades were determined by inductively coupled plasma-atomic emission spectroscopy (Ultima C, Horiba Jobin Yvon, Longjumeau, France). Phytoliths were extracted following the 'wet digestion'-protocol detailed in Table 2 of Corbineau et al. (2013). The protocol involves treatment of the sample with different chemical agents (HCl, $H_2SO_4$, $H2O_2$, $HNO_3$) to remove organic and carbonate compounds. The pure phytolith concentrates were mounted on microscope slides in Canada Balsam and the morphological types were counted using light microscopy at a 600X magnification. The epidermal silicified intercoastal long cells were quantified relative to the silicified short cells to obtain information on the silicification process (Alexandre et al., 2019).

The phytolith samples (1.6 mg) were dehydrated at 1100 °C under a flow of $N_2$ (Chapligin et al., 2010) to prevent the formation of siloxane from silanol groups during dehydroxylation. Molecular $O_2$ was extracted using the IR laser-heating fluorination technique (Alexandre et al., 2006; Crespin et al., 2008; Outrequin et al., 2021). At the end of the procedure, the gas was passed through a -114 °C slush to refreeze any molecule interfering with the mass 33 (e.g., NF potentially remaining in the line). The gas was directly sent to a ThermoQuest Finnigan Delta V Plus dual-inlet mass spectrometer (Massachusetts, USA) for triple oxygen isotope analysis. Each gas sample was run twice with each run consisting of eight dual-inlet cycles. A third run was performed when the standard deviation on the first two averages was higher than 12 per meg for $^{17}O$-excess. The reproducibility for $\delta^{18}O$ and $^{17}O$-excess measurements of the quartz laboratory standard was 0.16 ‰ and 8 per meg, respectively (1 SD, n = 5). For the phytolith samples, the precision for $\delta^{18}O$ and $^{17}O$-excess was always better than 0.5 ‰ and 12 per meg (1 SD), respectively. The sample measurements were corrected using a quartz laboratory standard analyzed at the beginning of the day until a $^{17}O$-excess plateau was reached and again at the end of the day. The isotope composition of the

reference gas was determined against NBS28. For robust comparisons between silica and water isotope compositions, the phytolith data are normalized to VSMOW-SLAP scale (Outrequin et al., 2021).

## 2.4 Modelling

According to the C-G isotope steady state model (Craig and Gordon, 1965; Dongmann et al., 1974; Farquhar et al., 2007; Cernusak et al., 2016), the isotope ratio of the evaporated water pool in the leaf ($R_e$) is:

$$R_e = \alpha_{eq}\alpha_{diff}(1-h)R_S + \alpha_{eq}hR_V,  \tag{2}$$

where $R_V$ and $R_S$ denote the isotope ratios ($^2H/^1H$, $^{17}O/^{16}O$ and $^{18}O/^{16}O$) of atmospheric water vapor and source water, respectively. $h$ is the ratio of the actual vapor pressure in the atmosphere to the saturation vapor pressure inside the leaf (i.e. at leaf temperature, $T_{leaf}$). When the leaf-to-air temperature gradient is small, $h$ is equal to RH. The isotope fractionation during water vapor diffusion in air through the leaf stomata and boundary layer ($\alpha_{diff}$) was estimated as:

$$\alpha_{diff} = \frac{\alpha_{kin}/g_s + \alpha_{kin}{}^{2/3}g_b}{1/g_s + 1/g_b}  \tag{3}$$

where $g_s$ and $g_b$ (mol m$^{-2}$ s$^{-1}$) denote the stomatal and leaf boundary layer conductances, and $\alpha_{kin}$ denotes the kinetic isotope fractionation during molecular diffusion of water vapor in air. We took $^{18}\alpha_{kin} = 1.028$ and $^2\alpha_{kin} = 1.025$ from Merlivat et al. (1978) for $^{18}O/^{16}O$ and $^2H/^1H$, respectively. Stomatal and boundary layer conductances measured continuously on a single leaf using the LI-COR gas exchange system (see Section 2.1) are used for modelling. For equilibrium isotope fractionation between water and water vapor, temperature-dependent fractionation factors ($\alpha_{eq}$) for $^{18}O/^{16}O$ and $^2H/^1H$ reported by Majoube et al. (1971) are used herein. The fractionation factors for $^{17}O/^{16}O$ are derived from those of $^{18}O/^{16}O$ according to $^{17}\alpha = {}^{18}\alpha^\theta$ using $\theta_{eq} = 0.529$ for liquid-vapor equilibrium (Barkan and Luz, 2005) and $\theta_{kin} = 0.5185$ for the kinetic fractionation during molecular diffusion (Barkan and Luz, 2007).

The bulk grass leaf water at isotope steady state ($R_{leaf,ss}$) represents a mixture of an evaporated water pool in the lamina mesophyll whose isotope composition is predicted by the C-G model ($R_e$, Eq. (2)), and an unevaporated pool in the leaf veins and associated ground tissues, whose isotope composition matches $R_s$ (Leaney et al., 1985; Yakir et al., 1994; Hirl et al., 2019):

$$R_{leaf,ss} = (1-f)R_e + fR_s  \tag{4}$$

where $f$ represents the water volume fraction of the unevaporated pool and was set to 0.2 in our study. Similar values were used in previous studies on grass leaf water (Wang et al., 2018; Alexandre et al., 2019; Hirl et al., 2019). Instead of a mixing equation, the Péclet effect can be considered to estimate the bulk leaf water isotope composition (Farquhar and Lloyd, 1993; Farquhar et al., 2007; Holloway-Phillips et al., 2016):

$$R_{leaf,ss} = R_s + (R_e - R_s)\frac{1-e^{-p}}{p}  \tag{5}$$

With p [= EL/CD] the Péclet number, where L is the effective path length, E is the grass leaf transpiration rate, C is the molar density of liquid water ($55500 \text{ mol m}^{-3}$), and D is the diffusivity of water ($2.3 \cdot 10^{-9} \text{ m}^2 \text{ s}^{-1}$ at 25 ºC). One single value of L was applied for the data set and adjusted to fit the observed grass leaf water isotope composition.

When the steady state cannot be reached, non-steady state enrichment of bulk leaf water ($R_{\text{leaf,nss}}$) can be modelled using the following equation (Dongmann et al., 1974; Farquhar and Cernusak, 2005; Hirl et al., 2019):

$$R_{\text{leaf,nss}}(t_0 + \Delta t) = R_{leaf,ss}(t_0 + \Delta t) + (R_{leaf,nss}(t_0) - R_{leaf,ss}(t_0 + \Delta t))e^{-\frac{\Delta t}{\tau}}, \qquad (6a)$$

With $\qquad\qquad \tau = \frac{W\alpha_{eq}\alpha_{diff}}{gw_i} \qquad\qquad\qquad\qquad\qquad (6b)$

where $g = g_s g_b/(g_s+g_b)$, $w_i$ is the mole fraction of water vapor in air in the intercellular spaces, $W$ is the leaf water content and $R_{\text{leaf,ss}}$ denotes the isotope composition of bulk leaf water at steady state, as predicted by Eq. (4). Similar to Farquhar & Cernusak (2005) or Hirl et al. (2019), we neglected diurnal changes in $W$, which should result in only $\sim 3$ % error in predicted leaf water isotope enrichment (Farquhar and Cernusak, 2005). We adjusted $W$ to fit the observed grass leaf water isotope composition. The best fit was found for $W$ of $6 \text{ mol m}^{-2}$.

Both steady state and non-steady state model calculations were performed for isotope ratios ($^2\text{H}/^1\text{H}$, $^{17}\text{O}/^{16}\text{O}$ and $^{18}\text{O}/^{16}\text{O}$) independently, and the secondary isotope parameters (d-excess and $^{17}$O-excess) were derived from predicted primary isotope values ($\delta^{17}\text{O}$, $\delta^{18}\text{O}$, $\delta^2\text{H}$) using the equations given in Section 1.

## 3 Results

### 3.1 Changes in the isotope composition of atmospheric water vapor, precipitation, and irrigation water

Over the experimental period, the isotope composition of irrigation water that mainly fed the soil water, was stable, averaging $-7.4 \pm 0.2$ ‰ for $\delta^{18}\text{O}$, $-48.5 \pm 0.7$ ‰ for $\delta^2\text{H}$, $10.7 \pm 0.6$ ‰ for d-excess and $31 \pm 6$ per meg for $^{17}\text{O}$-excess (Fig. A2). These values are close to the amount-weighted annual averages of precipitation in 2021: $-8.1 \pm 2.9$ ‰ for $\delta^{18}\text{O}$, $-52 \pm 24$ ‰ for $\delta^2\text{H}$, $12.0 \pm 3.5$ ‰ for d-excess and $29 \pm 11$ per meg for $^{17}\text{O}$-excess (Table S4). The precipitation ($730 \text{ mm a}^{-1}$) was mainly distributed between two periods in spring (April to May) and autumn (October to December) (Fig. 1, Table S4).

The annual average isotope composition of atmospheric water vapor was $-17.4 \pm 3.1$ ‰ for $\delta^{18}\text{O}$, $-126 \pm 24$ ‰ for $\delta^2\text{H}$, $13.0 \pm 1.7$ ‰ for d-excess and $28 \pm 5$ per meg for $^{17}\text{O}$-excess. These values coincide with $\delta^{18}\text{O}$, $\delta^2\text{H}$, d-excess and $^{17}\text{O}$-excess values estimated for a water vapor in isotope equilibrium with the amount-weighted precipitation (Table S4). As for precipitation, the atmospheric water vapor monthly averages in $\delta^{18}\text{O}$ and $\delta^2\text{H}$ increase from winter to summer, whereas averages in d-excess and $^{17}\text{O}$-excess decrease (Fig. 1; Table S4). During the 24-hour monitoring, $\delta^{18}\text{O}$ of atmospheric water

vapor increased overnight from about -16 to -12 ‰ and then stabilized. The d-excess and $^{17}$O-excess of atmospheric water vapor showed diurnal variations, reaching respective minimum values of -3.2 ‰ and -10 per meg in the early morning and respective maximum values of 18.4 ‰ and 36 per meg at noon (Table S5).

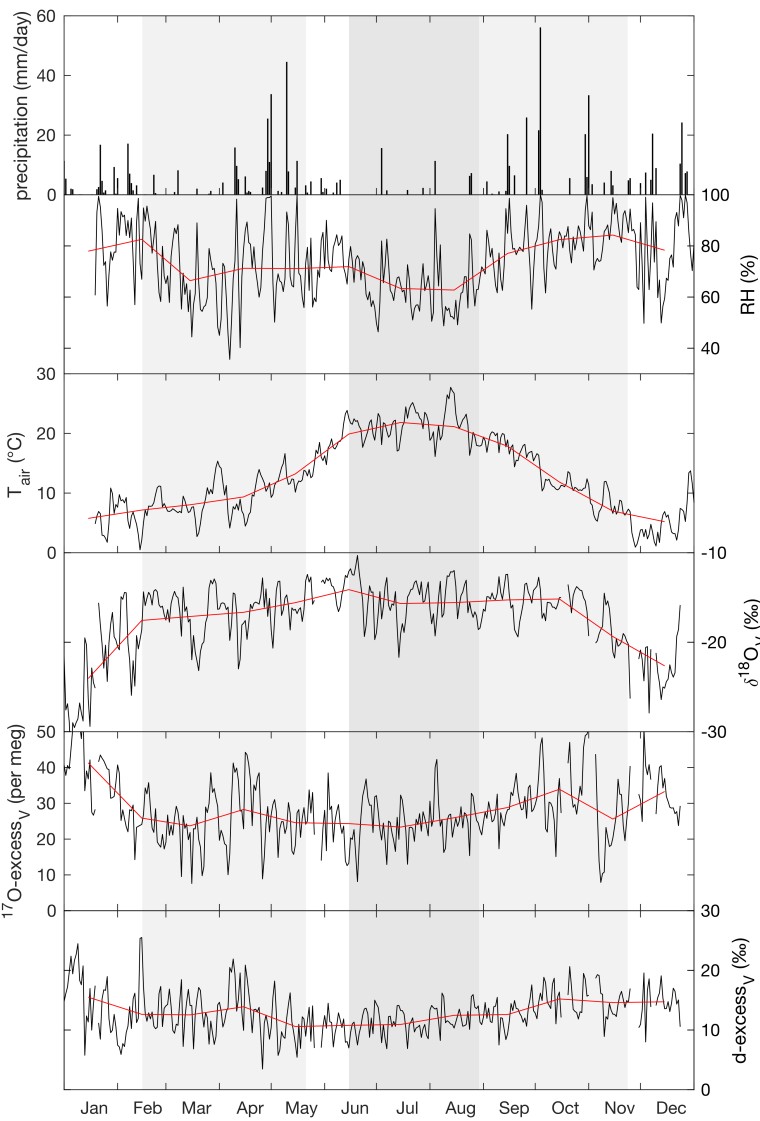

315

**Figure 1:** Daily precipitation amount, daily (black) and monthly (red) means of relative humidity (RH) and atmospheric temperature ($T_{air}$) measured at 60 cm above the ground next to the grass plot, and the isotope composition of atmospheric water vapor ($\delta^{18}O_V$, $^{17}$O-excess$_V$, d-excess$_V$) measured at 40 cm height above the grass plot monitored at the O₃HP platform from February to November 2021. The three regrowth periods lasting from 17 February–20 May 2021 (spring), from 15 June–

27 August 2021 (summer) and from 27 August–23 November 2021 (autumn) are indicated by shaded areas.

**3.2 Changes in RH, temperature, stomatal conductance, transpiration, and the isotope composition of grass leaf water**

Table 1 and Figure 2 show changes in RH, h, $T_{air}$, $T_{plot}$, *F. arundinacea* leaf transpiration and stomatal conductance averaged over 30 minutes before the 8 grass leaf samplings at midday. RH is always equal or lower than h (by less than 9 %) but co-varies with h from low values in spring and summer (30–40 %) to high values in autumn (ca. 64 %). $T_{plot}$ is 1–3 ℃ lower than $T_{air}$ but changes along with $T_{air}$ from a measurement day to another, with high values in summer (ca. 25 °C), and lower values in spring (ca. 18 °C) and autumn (ca. 14 °C). Figure A3 shows five daily variations of $T_{air}$, $T_{plot}$ and $T_{leaf}$. Although $T_{leaf}$ varies spatially within the plot, its spatial average around midday is close to $T_{plot}$ (Fig. A3), supporting that $T_{plot}$ can be considered as an approximation of $T_{leaf}$. Transpiration and stomatal conductance are relatively stable from a measurement day to another, varying from 1.1–3.7 mmol m$^{-2}$ s$^{-1}$ and 50–130 mmol m$^{-2}$ s$^{-1}$, respectively (Fig. 2).

The isotope composition of *F. arundinacea* leaf water sampled at midday is also shown in Table 1 and Figure 2. The grass leaf water has $\delta^{18}O$ (-0.05 ‰ to 20.1 ‰) and $\delta^2H$ (-31 ‰ to 18 ‰) that are higher than irrigation water, and d-excess (-31.0 ‰ to -142.8 ‰) and $^{17}O$-excess (17 per meg to -165 per meg) that are lower than irrigation water, as can be expected for an evaporation signal. The changes in $\delta^{18}O$, $\delta^2H$, d-excess and $^{17}O$-excess observed from a sampling day to another follow the changes in RH and h (Fig. 2). Evaporative isotope enrichment is highest in May and July when RH is low and lowest in November when RH is high. Samples from October and November have similar d-excess as expected from little variation in RH (64 ± 2 %). However, their $^{17}O$-excess values differ by 66 per meg. The reason for this difference in $^{17}O$-excess remains unclear.

Table 1 and Figure 3 show the 24-hour evolution of the isotope composition of grass leaf water from 14–15 June 2021 in relation to RH, h, $T_{air}$, $T_{plot}$, *F. arundinacea* transpiration and stomatal conductance. $T_{air}$ and RH range from 14 °C to 31 °C and 38 % to 97 %, respectively. $T_{plot}$ is ca. 1 °C higher than $T_{air}$ at night, and up to 4 °C lower than $T_{air}$ during daytime. During daytime, stomatal conductance measured continuously on a single leaf, ranges from 60 to 120 mmol m$^{-2}$ s$^{-1}$ and co-varies with transpiration (1.3–3.9 mmol m$^{-2}$ s$^{-1}$). However, stomatal conductance varies greatly (by 200–500 mmol m$^{-2}$ s$^{-1}$) between different leaves in the grass plot (Table S1, Fig. A4). At night, stomatal conductance is never higher than 20 mmol m$^{-2}$ s$^{-1}$, while transpiration remains lower than 0.5 mmol m$^{-2}$ s$^{-1}$. The isotope variability of grass leaf water on this diurnal scale is of the same order of magnitude as the changes observed among samples collected at midday in different months. The evolution of the isotope composition of grass leaf water follows RH and h, except for samples collected at night and in the early morning when transpiration is low. During this time, stomatal closure impeded exchange between the leaf and the atmosphere, decoupling the isotope composition of grass leaf water from RH.

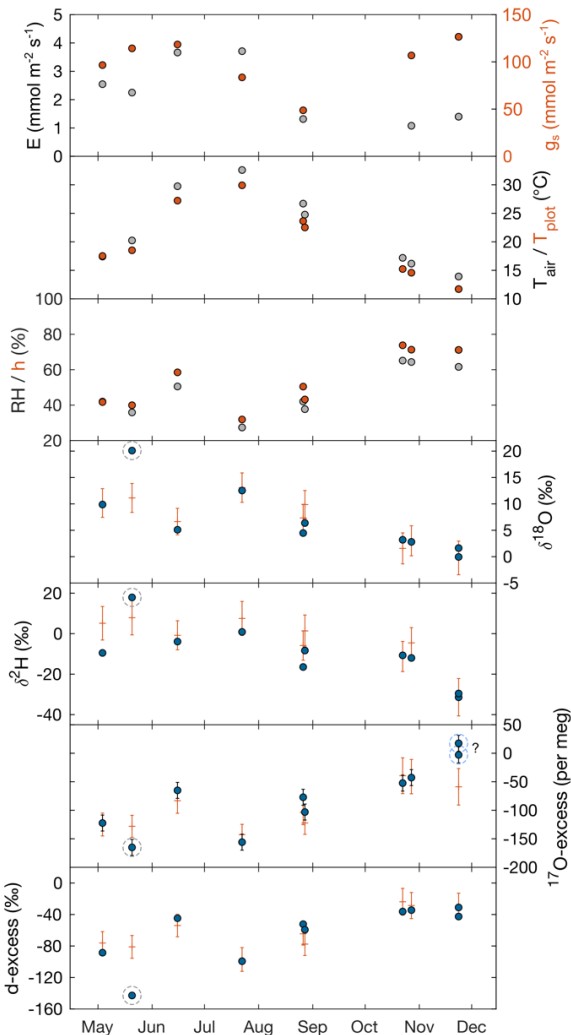

**Figure 2:** *F. arundinacea* transpiration (E) and stomatal conductance ($g_s$) measured on a single leaf blade using the LI-COR gas exchange system, atmospheric temperature ($T_{air}$), plot-scale grass leaf temperature ($T_{plot}$), relative humidity (RH), water vapor pressure ratio between leaf and the atmosphere (h) and measured (circles) and predicted (+) isotope composition of *F. arundinacea* leaf water ($\delta^{18}O$, $\delta^2H$, $^{17}O$-excess, d-excess) for midday samples over the year 2021 (see Table 1 for sampling dates). Error bars of isotope data represent analytical precision (see method section). The modeled isotope composition of bulk grass leaf water is predicted by the C-G steady state model combined with the mixing equation (Eq. (4)) using average environmental conditions over 30 minutes before sampling (Table 1, S5). The model uncertainty (1 SD) was estimated using a Monte Carlo simulation accounting for uncertainty of input variables (RH ± 1 %, $T_{plot}$ ± 2 °C, $\delta^{18}O_S$ ± 0.2 ‰, $\delta^2H_S$ ± 0.7 ‰, d-excess$_S$ ± 0.6 ‰, $^{17}O$-excess$_S$ ± 6 per meg, $\delta^{18}O_V$ ± 0.2 ‰, $\delta^2H_V$ ± 1.8 ‰, d-excess$_V$ ± 0.9 ‰, $^{17}O$-excess$_V$ ± 14 per meg, $g_s$ ± 100 mmol m$^{-2}$ s$^{-1}$, and the fraction of unevaporated water pools (*f*) ± 0.1). Gray dashed circles indicate the sample that has been likely affected by evaporation during sampling. Light blue dash circles indicate samples with anomalously high $^{17}O$-excess relative to d-excess.

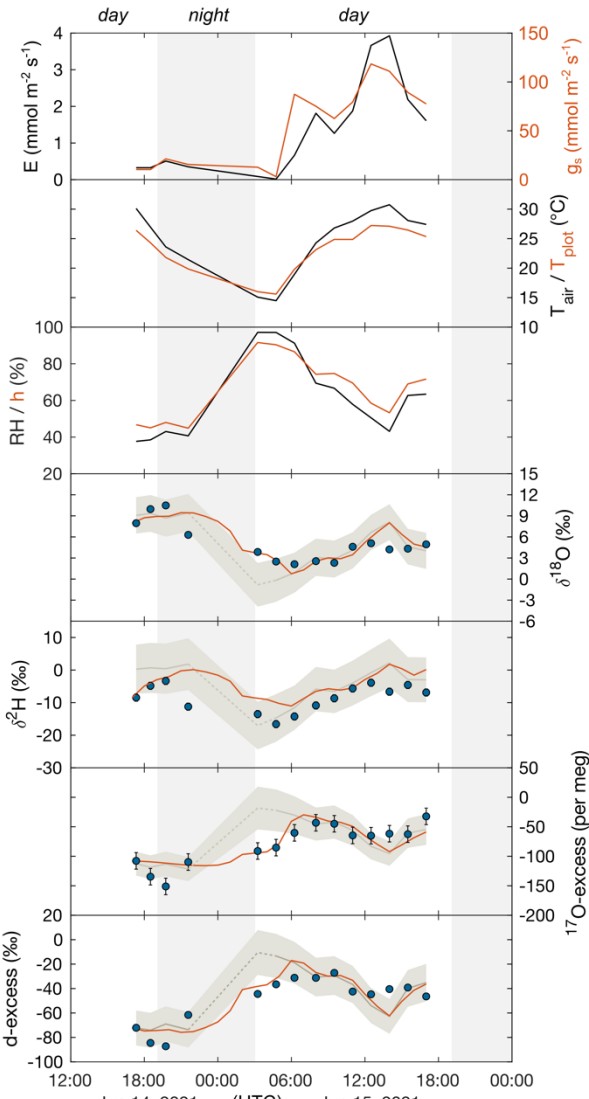

**Figure 3:** 24-hour monitoring of *F. arundinacea* transpiration (E) and stomatal conductance ($g_s$) measured on a single leaf blade using the LI-COR gas exchange system, atmospheric temperature ($T_{air}$), plot-scale grass leaf temperature ($T_{plot}$), relative humidity (RH), water vapor pressure ratio between leaf and the atmosphere (h), and the observed (circles) and predicted steady state (pale gray curve, Eq. (4)) and non-steady state (orange curve, Eq. (6a)) isotope composition of *F. arundinacea* leaf water ($\delta^{18}O$, $\delta^2H$, $^{17}O$-excess, d-excess) from 14–15 June 2021. Error bars of isotope data represent analytical precision (see method section). Shaded areas mark nighttime intervals. The isotope composition of grass leaf water is predicted using average environmental conditions over 30 minutes before sampling (Table 1, S5, S6). The pale gray shaded area represents model uncertainty (1 SD) of the predicted steady-state leaf water isotope composition estimated using a Monte Carlo simulation (see caption Figure 2). The dashed part of the steady state prediction represents the time when grass leaf water isotope composition deviates from steady state due to low transpiration and long leaf water residence times (see discussion for details).

### 3.2.1 Model-data comparison

For six of eight midday samplings, the isotope composition of bulk grass leaf water predicted by the C-G steady state model combined with the mixing equation (Eq. (4)) using boundary conditions averaged over 30 minutes before sampling (Table S5) agrees with the measured isotope values within model uncertainty, that is in average $\pm 2.8$ ‰, $\pm 7.9$ ‰, $\pm 15$ ‰, and $\pm 24$ per meg for $\delta^{18}O$, $\delta^2H$, d-excess and $^{17}O$-excess, respectively (Fig. 2). Samples collected on 20 May 2021 and 23 November 2021 show larger discrepancies between observed and predicted values. The May sample has significantly

higher $\delta^{18}O$ (> 8 ‰) and $\delta^2H$ (> 10 ‰), and lower d-excess (59 ‰) and $^{17}O$-excess (33 per meg) than respective steady state values predicted by the two-pool mixing model (Eq. (4)) (Fig. 2). These large deviations are indicative of stronger evaporation than expected. In view of the large magnitude of the deviation, we suppose that this sample was affected by evaporation during sampling. We therefore exclude this sample from further discussion. For the November sample, $\delta^{18}O$, $\delta^2H$ and d-excess agree within 1.1 ‰, 1 ‰ and 8 ‰ with the predicted steady state values, respectively. However, the $^{17}O$-excess is 66 per meg higher

than the predicted steady state value (Fig. 2). The reason for this discrepancy remains unclear.

For the 24-hour monitoring, the C-G steady state model combined with the two-pool mixing equation (Eq. (4)) reproduces the evolution of the isotope composition of grass leaf water during the day, but not at night and in the early morning, when stomatal conductance and transpiration are low (Fig. 3). During daytime, best agreement between predicted and observed grass leaf water is found for samples collected on the morning of 15 June 2021 until midday, with deviations lower than $\pm 0.6$ ‰ for

$\delta^{18}O$, $\pm 5$ ‰ for $\delta^2H$, $\pm 6$ ‰ for d-excess and $\pm 8$ per meg for $^{17}O$-excess. However, on the afternoon of 15 June 2021, when transpiration is highest, observed $\delta^{18}O$ and $\delta^2H$ are 1.5–4 ‰ and 3–9 ‰ lower, and d-excess and $^{17}O$-excess are 9 ‰ and 34 per meg lower than predicted values, respectively. In contrast, on the evening of 14 June 2021, observed $\delta^{18}O$ are 1–2 ‰ higher, whereas $\delta^2H$, d-excess and $^{17}O$-excess are respectively 4–6 ‰, 18 ‰, and 38 per meg lower than respective steady state values predicted by the two-pool mixing model (Eq. (4)). The non-steady state equation (Eq. (6)) was applied for night

predictions to match the data (Fig. 3). Differences between predicted non-steady state and observed values at night range from 0.2–3.6 ‰ for $\delta^{18}O$, 5–12 ‰ for $\delta^2H$, 3–19 ‰ for d-excess and 1–31 per meg for $^{17}O$-excess (Table S6). Note that a grass leaf water content of 6 mol m$^{-2}$ is required for the model to fit the data (Table S6). This value is higher than leaf water contents reported for grasses in previous studies (2–4 mol m$^{-2}$; Hirl et al., 2019; Barbour et al., 2021).

### 3.2.2 Sensitivity tests

Figure 4 shows for the 24-hour monitoring the uncertainty of the bulk grass leaf water isotope composition predicted for steady state conditions (Eq. (4)) introduced by the precisions associated with the measurement of the main model parameters. A $\pm 5$ % uncertainty on RH introduces an uncertainty of $\pm 1.5$ ‰ on $\delta^{18}O$, $\pm 4.0$ ‰ on $\delta^2H$, $\pm 10$‰ on d-excess, and $\pm 13$ per meg on $^{17}O$-excess of grass leaf water (Fig. 4a–d). For an RH range of 40–80 %, an uncertainty of $\pm 0.1$ on the fraction of the unevaporated water pool ($f$) leads to an uncertainty of 2.2–0.8 ‰ on $\delta^{18}O$, 6–2 ‰ on $\delta^2H$, 12–4 ‰ on d-excess and 16–

6 per meg on $^{17}O$-excess (Fig. 4e–h). For the same RH range, misestimation of $\Delta T_{leaf-air}$ by 2 °C leads to an uncertainty of 1.3–

2.7 ‰ on $\delta^{18}O$, 1.5–5.1 ‰ on $\delta^{2}H$, 9–17 ‰ on d-excess and 11–29 per meg on $^{17}O$-excess (Fig. 4i–l). Assuming $T_{leaf}$ equals $T_{air}$, instead of measuring $T_{plot}$, increases the difference between the predicted and observed daytime $\delta^{18}O$, $\delta^{2}H$, d-excess and $^{17}O$-excess values by $1.1 \pm 1.2$ ‰, $2.4 \pm 0.5$ ‰, $5 \pm 11$ ‰ and $10 \pm 14$ per meg, respectively (Fig. 4i–l, orange curve). By contrast, assuming $T_{leaf}$ is 2 °C lower than $T_{air}$ only slightly increases the difference between predicted and observed daytime $\delta^{18}O$, $\delta^{2}H$, d-excess and $^{17}O$-excess values by $0.2 \pm 0.6$ ‰, $3.0 \pm 5.5$ ‰, $2 \pm 4$ ‰ and $3 \pm 5$ per meg, respectively (Fig. 4i–l, light blue curve). In contrast to RH, $f$ and $\Delta T_{leaf\text{-}air}$, measurement uncertainties on the isotope composition of the source water (irrigation water) and atmospheric water vapor, introduce uncertainties on the isotope composition of grass leaf water that are close to or lower than analytical precision (Fig. 4m–t). Using the isotope composition of atmospheric water vapor estimated from isotope equilibrium with the mean annual amount-weighted $O_3HP$ precipitation (Table S4) instead of measured values, increases the difference between predicted and observed daytime $\delta^{18}O$, $\delta^{2}H$, d-excess and $^{17}O$-excess values by $1.2 \pm 2.0$ ‰, $12.5 \pm 12.3$ ‰, $0 \pm 2$ ‰ and $3 \pm 8$ per meg, respectively (Fig. 4q–t, light blue curve). Observed spatial variability of stomatal conductance of up to 500 mmol m$^{-2}$ s$^{-1}$ introduces a bias on the $\delta^{18}O$, $\delta^{2}H$, d-excess and $^{17}O$-excess of less than 0.5 ‰, 0.5 ‰, 3.5 ‰ and 10 per meg, respectively (Fig. 4u–x).

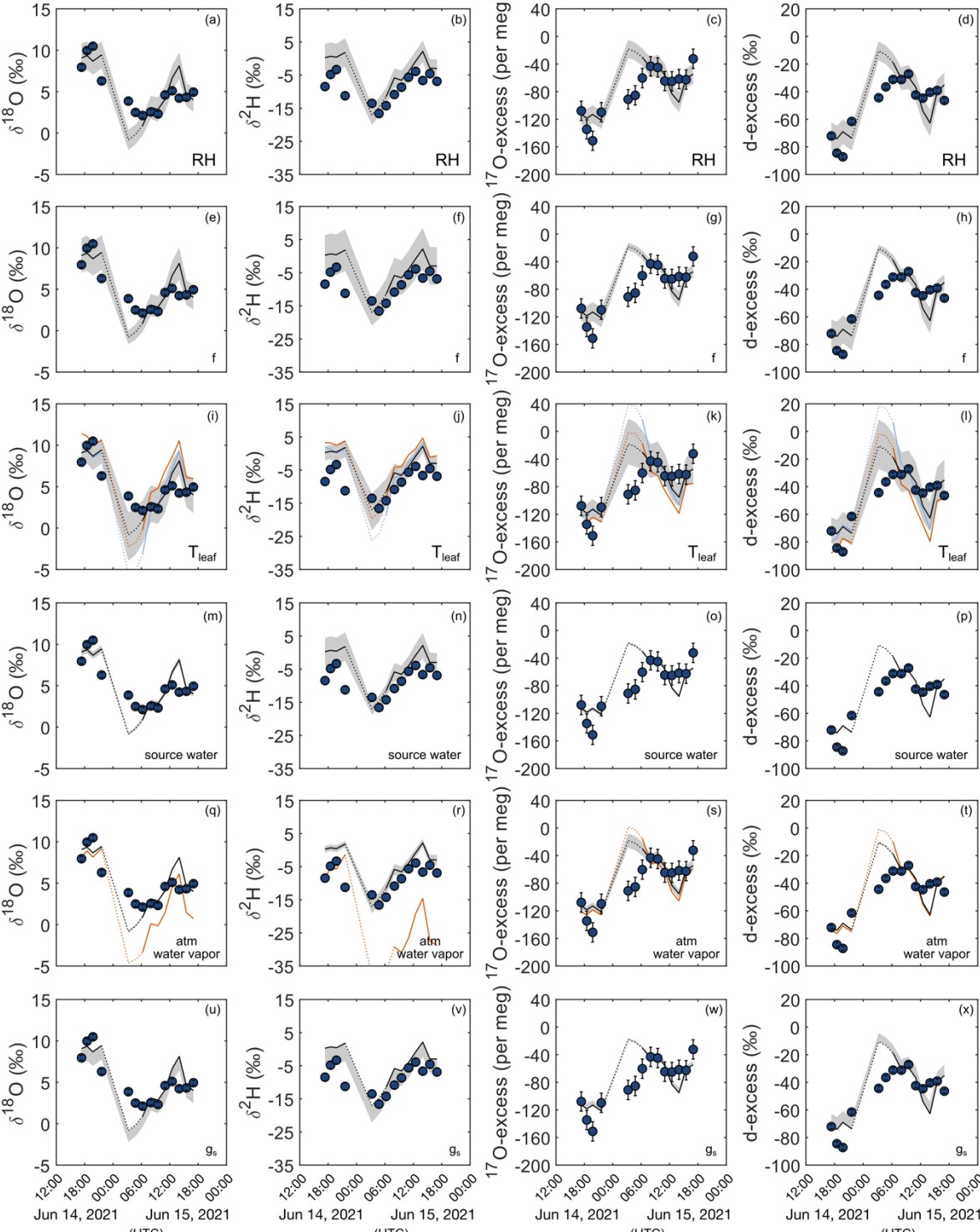

**Figure 4:** Sensitivity of $\delta^{18}O$, $\delta^2H$, $^{17}O$-excess and d-excess of leaf water to changes in environmental and plant physiological parameters. Green circles represent measured *F. arundinacea* leaf water isotope composition over a 24-hour period from 14–15 June 2021. The black line shows the steady state leaf water isotope composition predicted by the C-G steady state model combined with the mixing equation (Eq. (4)) using mean boundary conditions over 30 minutes before sampling (Table 1). Shaded areas indicate the sensitivity of the predicted leaf water isotope composition for relative humidity (RH) ($\pm 5$ %) (a–d), the fraction of unevaporated water pools (*f*) ($\pm 0.1$) (e–h), leaf temperature ($T_{leaf}$) ($\pm 2$ °C) (i–l), the isotope composition of source water ($\pm 0.2$ ‰ for $\delta^{18}O_S$, $\pm 0.7$ ‰ for $\delta^2H_S$, $\pm 0.6$ ‰ for d-excess$_S$, $\pm 6$ per meg for $^{17}O$-excess$_S$) (m–p), the isotope composition of atmospheric water vapor ($\pm 0.2$ ‰ for $\delta^{18}O_V$, $\pm 1.8$ ‰ for $\delta^2H_V$, $\pm 0.9$ ‰ for d-excess$_V$, $\pm 14$ per meg for $^{17}O$-excess$_V$) (q–t) stomatal conductance ($g_s$) ($\pm 100$ mmol m$^{-2}$ s$^{-1}$) (u–x). Coloured curves show the isotope composition of leaf water predicted by the C-G steady state model combined with the mixing equation (Eq. (4)) (i) when assuming leaf temperatures being equal to atmospheric temperature (panel i–l, orange), (ii) when assuming leaf temperatures being 2 °C lower than atmospheric temperature (panel i–l, light blue), and (iii) when estimating atmospheric water vapor from isotope equilibrium with source water (irrigation water) (panel q–t, orange).

### 3.3 Changes in climate averages, grass height, silicification rate, and triple oxygen isotope composition of phytolith assemblages

Table 2 shows daily and daytime climate averages, maximum grass height, silicification rate, ratio of long cell to short and long cell phytoliths, and the triple oxygen isotope composition of phytoliths for the three regrowth periods. Daily average $T_{air}$ is from 9 °C to 22 °C and daily average RH is from 64 % to 81 %. Daytime average $T_{air}$ is about 2.4 °C higher than the daily average. Daytime average RH is about 8 % lower than the daily average. Daily averages of $T_{plot}$ are similar to $T_{air}$, so that RH approximates h (cf. Section 2.4). During daytime, averages of RH and h differ by 1–4 % due to $\Delta T_{leaf\text{-}air}$ ranging from -1.1 °C to 0.3 °C. Daytime average h is 61 % in spring and summer, and 76 % in autumn. The average soil water content is always higher than $0.20 \pm 0.05$ L L$^{-1}$, whatever the regrowth, supporting that the grass plot is always well-watered, and that water stress is excluded.

Grass height increases exponentially during spring regrowth, and linearly during summer regrowth (Fig. A5). During the autumn regrowth, the grass height increases only in the first month of the regrowth and stabilizes thereafter (Fig. A5). The silicification rate (from 2.7 to 5.9 SiO$_2$ g$^{-1}$ d$^{-1}$), and the proportion of silicified long cell phytoliths (from 30 to 70 % of short and long cells) increase with the number of regrowth periods, without any correlation with RH or h that varied little from a regrowth to another (Table 2). The $\delta'^{18}O$ and $^{17}O$-excess of the grass leaf phytoliths are similar in spring and summer ($36.2 \pm 0.5$ ‰ and $-260 \pm 5$ per meg, respectively; Table 2) and slightly different in autumn (34.3 ‰ and -234 per meg, respectively). These isotope values fall within the range of values observed in previous growth chamber calibrations (Alexandre et al., 2018, 2019; Outrequin et al., 2021). The $^{17}O$-excess$_{phyto}$ coincide with the lower range of values reported for phytoliths extracted from soils in Western and Central Africa (Alexandre et al., 2018).

### 3.4 Relationship between the [17]O-excess of grass phytoliths and leaf water

Numerous studies have investigated the temperature-dependent isotope fractionation between amorphous and/or biogenic silica and their formation water ($^{18}\alpha_{phyto-H2O}$) with variable results (e.g., O'Neil and Clayton, 1964; Knauth and Epstein, 1976; Shemesh et al., 1992; Brandriss et al., 1998; Hu and Clayton, 2003; Dodd, 2011, and many more). Here we use temperature-dependent $^{18}\alpha_{phyto-H2O}$ obtained for the diatom-water pair by Dodd and Sharp (2010) (1.0326 at 20°C). The triple oxygen isotope exponent between silica and water ($\theta_{phyto-H2O}$) linking $^{17}\alpha_{phyto-H2O}$ to $^{18}\alpha_{phyto-H2O}$ ($^{17}\alpha = {}^{18}\alpha^{\theta}$), has been defined as 0.524 ± 0.0002 for the 5–35°C temperature range (Cao and Liu, 2011; Sharp et al., 2016). However, a different value of 0.522 ± 0.001 was consistently obtained for phytoliths, reproducible regardless of bio-climatic constraints (Outrequin et al., 2021). Using this apparent $\lambda_{phyto-H2O}$, we calculated the triple oxygen isotope compositions of the formation water (FW) in equilibrium with the phytolith samples obtained from the three regrowths (Fig. 5). The reconstructed triple oxygen isotope composition of FW is close to that estimated for daytime average climate conditions of the three regrowths using the C-G model combined with the mixing equation (Eq. (4)) (Fig. 5). The differences are lower than 1.8 ‰ and 33 per meg for $\delta'^{18}O$ and $^{17}O$-excess, respectively. Using the same $^{18}\alpha_{phyto-H2O}$, but $\lambda_{phyto-H2O}$ of 0.524, the $^{17}O$-excess of FW is largely underestimated by 35–60 per meg compared to model predictions (Fig. 5).

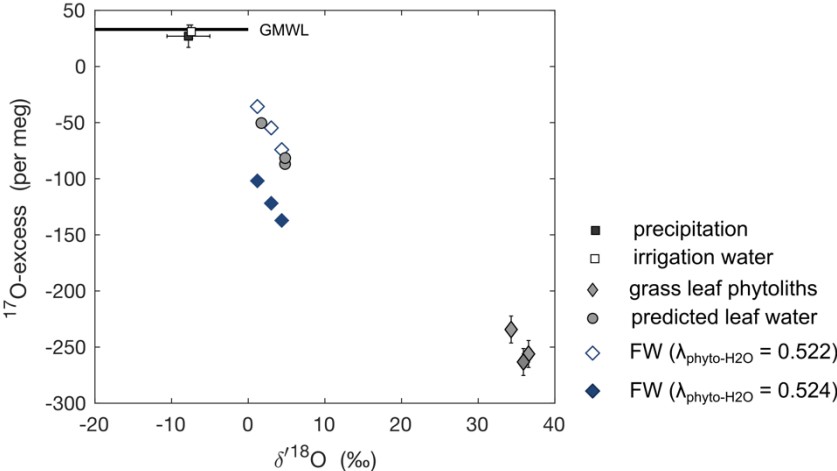

**Figure 5:** $^{17}O$-excess vs $\delta'^{18}O$ of amount-weighted annual average precipitation, average irrigation water, and the measured isotope composition of phytoliths extracted from *F. arundinacea* grass leaves harvested on 20 May 2021 (spring), 27 August 2021 (summer), and 23 November 2021 (autumn). Also shown are the formation water (FW) predicted using temperature-dependent $^{18}\alpha_{SiO2-H2O}$ from Dodd and Sharp (2010) and $\lambda_{phyto-H2O}$ of 0.522 or 0.524, and the isotope composition of bulk leaf water predicted by the C-G model for steady state conditions combined with the mixing equation (Eq. (4)) using average daytime boundary conditions for the three regrowth periods (Table 2). Error bars represent analytical precisions (see methods section), except for precipitation, for which the amount-weighted standard deviation is indicated.

## 4 Discussion

### 4.1 Parameters responsible for discrepancies between observed and predicted isotope compositions of grass leaf water

Overall agreement between the observed and predicted leaf water $\delta^{18}O$ and $^{17}O$-excess trends from a sampling day to another shows that the C-G steady state model combined with the two-pool mixing equation (Eq. (4)) is appropriate for estimating seasonal scale variations in the triple oxygen isotope composition of grass leaf water at midday. The two-pool mixing model also correctly reproduces the trends in triple oxygen isotope evolution of leaf water during daytime, although observed and predicted values diverge little when transpiration is maximal in the early afternoon (Fig. 3). As shown by the sensitivity tests, $\Delta T_{leaf-air}$ contributes largely to model uncertainty (Fig. 4). Assumptions on $T_{leaf}$ equal to $T_{air}$ can explain the discrepancies between predicted and observed isotope values often reported in the literature. In the present case, $T_{leaf}$ was indirectly measured using $T_{plot}$ and large misestimation of $T_{leaf}$ (>2 °C) is unlikely. Part of the small model-data discrepancies in the afternoon on 15 June 2021 can result from RH measured at 60 cm above the ground next to the grass plot being lower than RH surrounding the grass leaf canopy, due to intense soil evaporation. Another bias may come from misestimation of the unevaporated water pool $f$ that can drive large variations in the triple oxygen isotope composition of leaf water, as shown by the sensitivity tests. The value of 0.2 chosen for $f$ in the present study is at the lower limit of previously reported values selected for grass species (0.2–0.4; Hirl et al., 2019; Barbour et al., 2021). Considering a value for $f$ of 0.4 instead of 0.2 would minimize the discrepancy between observed and predicted $\delta^{18}O$ of leaf water for the samples taken in the afternoon on 15 June 2021 (Fig. 4). Some studies suggested that $f$ may increase with increasing transpiration, due to increased advection of unevaporated xylem water, known as the Péclet effect (Farquhar and Lloyd, 1993; Cuntz et al., 2007). In contrast to a recent isotope study that found no evidence for the Péclet effect in grass leaves (Hirl et al., 2019), the data from the 24-hour monitoring shows a significant positive correlation ($R^2 = 0.49$) between transpiration and the difference between observed and predicted $\delta^{18}O$ values of leaf water. Considering the Péclet effect (Eq. (5)) instead of a simple mixing significantly reduces model-data discrepancies by 50–80 % and leads to deviations between predicted and observed $\delta^{18}O$ and $^{17}O$-excess of grass leaf water in the afternoon on 15 June 2021 that are lower than 1.1 ‰, and 12 per meg, respectively (Table S5). The Péclet effect can thus explain that the observed triple oxygen isotope composition of leaf water varies less than predicted when transpiration is high.

In agreement with previous studies on $\delta^{18}O$ and $\delta^2H$ (Farquhar and Cernusak, 2005; Cernusak et al., 2016), a non-steady state model is used to reproduce the trends in isotope evolution of leaf water at night when stomatal conductance and transpiration are low. Our results confirm the applicability of this model for the triple oxygen isotope composition of leaf water. In addition, the model-data comparison shows the advantage of $^{17}O$-excess over d-excess in detecting isotope non-steady state in leaf water on a diurnal scale. Figure 6a illustrates the $^{17}O$-excess vs $\delta'^{18}O$ evolution of leaf water from the beginning to the end of the night when transpiration is too low to reach the isotope steady state. RH of $96 \pm 2$ % persisting between 3:00 and 7:00 (LT) on 15 June 2021 drives the theoretical isotope steady state values to the upper end of the predicted trend on Figure 6a. However, due to the long leaf water residence time, the observed leaf water isotope composition evolves only slowly towards these

values without reaching them. This is well captured by the concave curvature of the non-steady state prediction (Fig. 6a). In contrast, linearity of evaporation trends in the d-excess vs $\delta^{18}O$ space challenges the differentiation between isotope steady state and non-steady state conditions, as illustrated in Figure 6b.

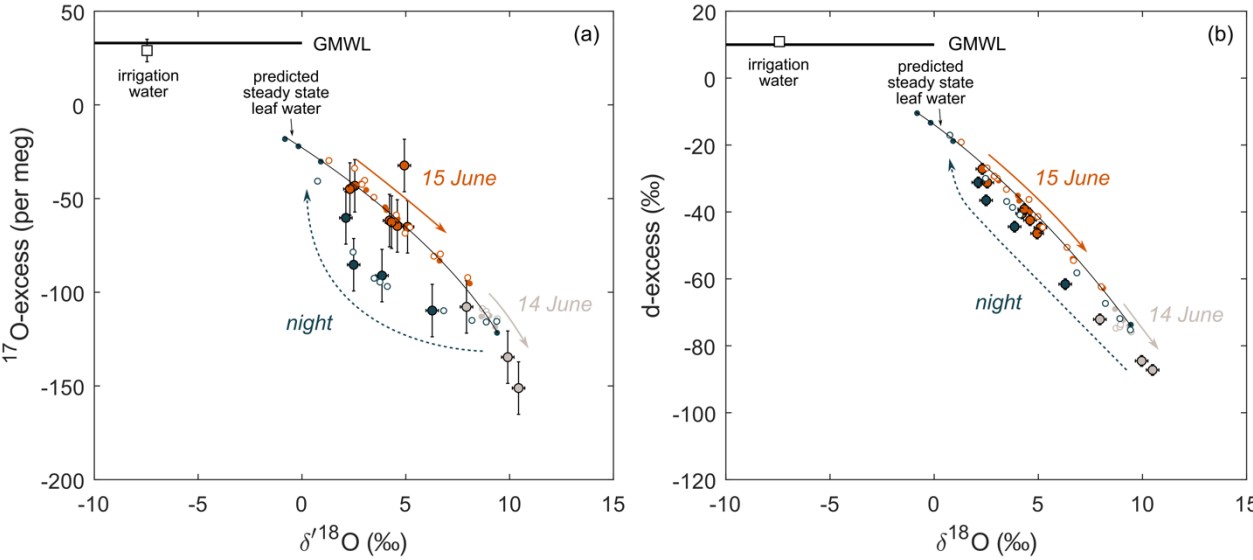

**Figure 6:** Comparison of predicted (small circles) and observed (large circles) *F. arundinacea* leaf water over a 24-hour period from 14–15 June 2021 in diagrams of (a) $^{17}O$-excess vs $\delta'^{18}O$ and (b) d-excess vs $\delta^{18}O$. Filled circles indicate the steady state model prediction (Eq. (4), Table S5), open circles indicate the non-steady state model prediction (Eq. (6), Table S6). Colours differentiate samples collected between 19:15 and 21:45 (LT) on 14 June 2021 (grey), between 14 June 2021 23:30 and 515 15 June 2021 08:15 (LT) (blue) and between 10:00 and 19:00 on 15 June 2021 (orange). The black line serves as a guide-of-the-eye for the trend in modelled isotope steady state values. The average isotope composition of the irrigation water over the experimental period is also shown. The global meteoric water line (GMWL) is shown for comparison.

### 4.2. What can we learn from measurements of $T_{plot}$ and triple oxygen isotope composition of atmospheric water vapor?

The sensitivity tests highlight the importance of plot-scale grass leaf temperature and the isotope composition of atmospheric 520 water vapor for accurate prediction of the isotope composition of leaf water.

The influence of variations in $T_{leaf}$ relative to $T_{air}$ on the isotope composition of leaf water is two-fold. On the one hand, changes in $T_{leaf}$ slightly modify the magnitude of equilibrium isotope fractionation at the liquid-vapor interface. A few degrees change in $T_{leaf}$ is however of minor importance for the isotope composition of leaf water. In contrast, changes in $\Delta T_{leaf\text{-}air}$, associated with changes in $T_{leaf}$, modify the water vapor pressure ratio between the leaf and the atmosphere, i.e. $h$. For example, a decrease 525 in $T_{leaf}$ from 20 to 18 ºC at constant $T_{air}$ of 20 ºC, modifies $h$ by 5–10 % for RH ranging from 40 to 80 %. As $h$ is the major driver of isotope variability in leaf water, even little variations in $\Delta T_{leaf\text{-}air}$ can therefore significantly influence the isotope composition of leaf water (Fig. 4i–l).

Accurate measurement of $T_{leaf}$ on plot-scale is challenging, as $T_{leaf}$ can vary considerably in space and time, particularly according to soil moisture, leaf transpiration, canopy structure and position, net radiation, elevation, and latitude (Still et al., 2019). Sufficient soil moisture supports transpiration, which generally leads to leaf cooling, i.e. $T_{leaf}$ lower than $T_{air}$. On the contrary, water stress is compensated by stomata closure, which stops transpiration and increases $T_{leaf}$. In this case, $T_{leaf}$ may exceed $T_{air}$, as demonstrated for irrigated *vs* rain-fed crops (Siebert et al., 2014). The amplitude of $\Delta T_{leaf-air}$ also increases with leaf size (Leuzinger and Körner, 2007). $\Delta T_{leaf-air}$ lower or equivalent to -2 °C was reported, at the ecosystem scale, for tropical forests (Rey-Sánchez et al., 2016), grasslands or cold desert areas, whereas larger differences were reported for cold forests and warm desert areas (Blonder and Michaletz, 2018). In the present case, continuous irrigation of the grass plot sustained the transpiration, leading to a daytime $T_{leaf}$ consistently near or below the daytime $T_{air}$ (Figs. A3, A6). However, under natural conditions, estimation of $T_{leaf}$ 2 °C lower than or equal to $T_{air}$ may lead to significant bias in modeled leaf water isotope composition. Figure A3 shows that $T_{plot}$ can be used to estimate $T_{leaf}$. The measurement of $T_{plot}$ using IR radiometry as performed here is easy to set up and is strongly recommended if high accuracy is sought in the estimate of $T_{leaf}$ at plot scale.

The isotope difference between source water and the atmosphere is another key determinant of the leaf water isotope composition. According to the C-G model (Eq. (2)), the influence of atmospheric water vapor relative to source water becomes increasingly important with increasing h (or RH). While the isotope composition of source water can be often constrained by measurements, accurate estimates of the isotope composition of atmospheric water vapor are difficult to obtain. In the absence of direct measurements, the $\delta^{18}O$ of atmospheric water vapor is often assumed to be in equilibrium with precipitation (e.g., Cernusak et al., 2002; Voelker et al., 2014; Bush et al., 2017; Li et al., 2017; Song et al., 2011; Flanagan and Farquhar, 2014). However, a recent comparison between modelled vapor and precipitation isotope compositions obtained from different isotope-enabled global climate models suggests that precipitation is rarely in isotope equilibrium with atmospheric water vapor (Fiorella et al., 2019). The deviation generally increases with increasing latitude. In continental areas, the $\delta^{18}O$ of near-surface atmospheric water vapor can be lower than suggested by isotope equilibrium with precipitation due to high evaporation fluxes from lakes (Krabbenhoft et al., 1990; Benson and White, 1994). Similarly, the $\delta^{18}O$ of atmospheric water vapor can be lower than suggested by isotope equilibrium, if precipitation is affected by sub-cloud re-evaporation, as has been reported for monsoon areas (Landais et al., 2010; Wen et al., 2010). Moreover, the equilibrium assumption is often not valid in semi-arid to arid regions, when precipitation is limited to a short period of the year and not representative for the annual average atmospheric conditions (Tsujimura et al., 2007; Voigt et al., 2021). The atmospheric water vapor record presented here supports the validity of the equilibrium assumption at the study site, for annual $\delta^{18}O$, $\delta^2H$, d-excess and $^{17}O$-excess averages. The agreement remains good at the monthly scale, but significant discrepancies occur for d-excess and $^{17}O$-excess during the summer months when RH is the lowest. Sub-cloud re-evaporation of precipitation can be invoked to explain the low d-excess and $^{17}O$-excess in summer precipitation, whereas d-excess and $^{17}O$-excess of atmospheric water vapor remain stable. At the diurnal scale, primary isotope ratios of atmospheric water vapor can vary strongly, often deviating from the monthly equilibrium value. This can lead to significant model-data discrepancies (Fig. 4). On diurnal scale, $^{17}O$-excess and d-excess of

atmospheric water vapor generally agree with the monthly equilibrium water vapor at daytime, when transpiration is high, but significantly deviate at night and in the early morning. Notably, the variations in $^{17}O$-excess of atmospheric water vapor over the 24-hour monitoring are low (45 per meg) compared to its large variability observed in leaf water (120 per meg) (cf. Table 1, S5). In comparison, $\delta^{18}O$ shows much higher variability in atmospheric water vapor (5 ‰) compared to leaf water (8 ‰) (cf. Table 1, S5).

### 4.3 Does the $^{17}O$-excess of grass leaf phytoliths reflect daily or daytime RH?

The triple oxygen isotope composition of bulk grass leaf phytoliths is influenced by their distribution along the leaf blade in relation to the leaf water isotope gradient and to silicification patterns (Alexandre et al., 2019; Outrequin et al., 2021). The triple oxygen isotope gradient along grass leaf blades can be predicted by a string-of-lakes model (Alexandre et al., 2019). However, the triple oxygen isotope composition of the bulk grass leaf water is independent of grass leaf length and predictable by the C-G model combined with the mixing equation (Eq. (4)) (Alexandre et al., 2019) or a Péclet effect. The bulk phytolith FW integrates the whole grass elongation period and is thus different from the sampled bulk leaf waters that only represent a snapshot in time. Short cells are among the first cell types to be silicified, sometimes even before the leaf transpires (Motomura et al., 2004; Kumar et al., 2017). The process is metabolically controlled and does not depend on the transpiration rate. Long cell silicification occurs in a second step in relation to transpiration (Motomura et al., 2004; Kumar et al., 2017). Moreover, in grass leaves, the epidermal cells are produced at the base of the leaf and pushed upward during the growth. Hence, epidermal cells along the leaf blade gather phytoliths that were formed at short and long distances relative to the leaf base, i.e. at isotopically low and high evaporative conditions, respectively. The combination of these two processes likely causes the apparent $\lambda_{phyto-H2O}$ being lower than the established $\theta_{SiO2-H2O}$ (= 0.524; Sharp et al., 2016) (Outrequin et al., 2021). The consistency of $\lambda_{phyto-H2O}$ equal to $0.522 \pm 0.001$ observed in this study and in previous calibrations (Outrequin et al., 2021), supports that the silicification patterns are systematic and similar for different climate conditions.

The relationship between $^{17}O$-excess$_{phyto}$ and RH was previously investigated in two growth chamber experiments where *F. arundinacea* was grown under different conditions of RH (40–60–80 %) and $T_{air}$ (20–24–28 °C) (Alexandre et al., 2018; Outrequin et al., 2021). The parameters were set constant for more than 10 days, without day-night cycles. Differences in $\delta^{18}O$ values between source water and atmospheric water vapor were set to 0 ‰ in the first experiment (Alexandre et al., 2018) and to 10 ‰ in the second experiment (Outrequin et al., 2021). The two equations obtained from these experiments were statistically similar (Outrequin et al., 2021). Linear regression through both datasets (n = 16) gives:

$$RH = 0.22 \, (\pm 0.01) \, ^{17}O\text{-excess}_{phyto} + 115.2 \, (\pm 3.9) \qquad (r^2 = 0.94) \qquad (7)$$

Here, under natural conditions, we investigate whether the RH obtained from Eq. (7) reflects daytime or daily conditions. RH values reconstructed from $^{17}O$-excess$_{phyto}$ obtained for the three regrowths applying Eq. (7) are closer to daytime averages (underestimation of RH by $4 \pm 4$ %) than to daily averages (underestimation of RH by $12 \pm 5$ %) (Fig. 7a, Table 2).

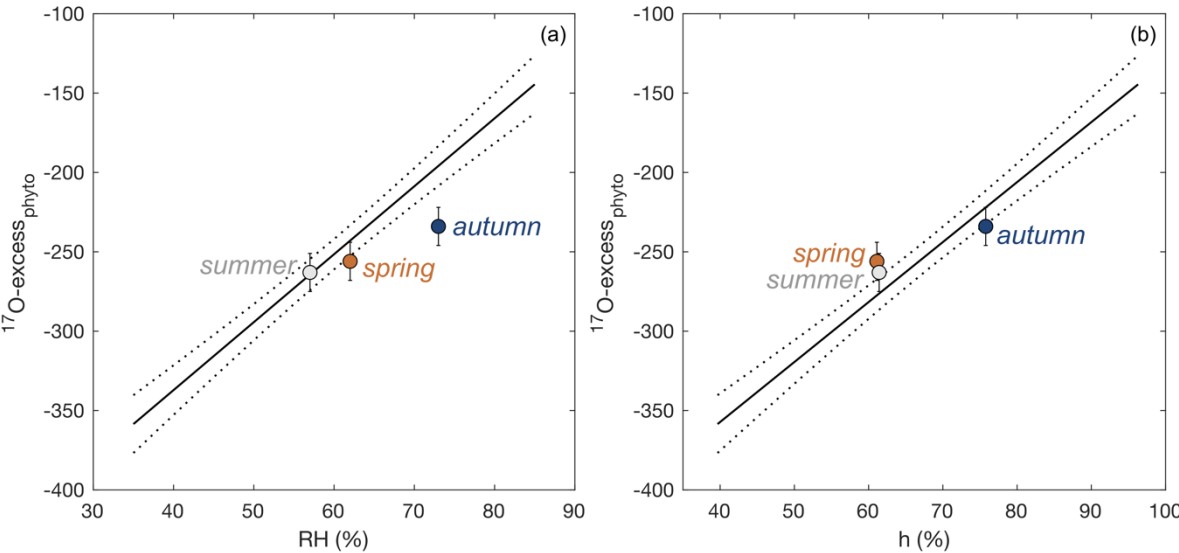

**Figure 7:** Observed $^{17}$O-excess$_{phyto}$ vs average daytime (a) relative humidity (RH), and (b) water vapor pressure ratio between the leaf and the atmosphere (h), for regrowth periods in spring, summer and autumn. The growth chamber calibration lines with 95 % confidence interval (Eqs. (6),(7)) are shown for comparison.

At night, low stomatal conductance and transpiration measured on *F. arundinacea* likely hamper the silicification due to cell water saturation relative to silica formation during daytime transpiration, explaining that daytime RH determines $^{17}$O-excess$_{phyto}$ (Fig. 7). Further, the low stomatal conductance of grasses observed at night causes its leaf water to deviate from isotope steady state. Hence, the $^{17}$O-excess of grass leaf water at night remains close to daytime values (Fig. 6). The low amount of phytoliths that may form overnight thus introduces little bias to the $^{17}$O-excess$_{phyto}$ of the phytolith sample. Nighttime stomatal conductance, however, can vary across biomes, depending among others on plant functional types and soil moisture (Tobin and Kulmatiski, 2018; Resco de Dios et al., 2019). A recent data compilation reported that tropical trees show the highest stomatal conductance at night, followed by desert species (Resco de Dios et al., 2019). The lowest stomatal conductance was found for non-tropical evergreen angiosperms including Mediterranean species. Therefore, for a given case, the magnitude of nighttime vs daytime transpiration must be assessed to determine whether the RH reconstructed from $^{17}$O-excess$_{phyto}$ reflects day and night or only daytime conditions.

RH estimated from $^{17}$O-excess$_{phyto}$ can be biased by variations in $\Delta T_{leaf-air}$. This is because the isotope composition of leaf water is not directly determined by RH, but rather the water vapor pressure ratio between the leaf and the atmosphere, i.e. h (cf. Eq. 2). As discussed in Section 4.2, $\Delta T_{leaf-air}$ of -2 ºC lead to h that are 5–10 % higher than RH. The calibration line obtained from growth chamber experiments is calibrated for $\Delta T_{leaf-air}$ of -2 ºC (Outrequin et al., 2021). The lower $\Delta T_{leaf-air}$ ranging from -1.1 ºC to 0.3 ºC observed in our study can explain the general underestimation of RH reconstructed from the calibration line (cf. Fig. 7a). The effect of $\Delta T_{leaf-air}$ can be removed when considering h instead of RH. We used the same datasets from

the growth chamber experiments as for RH (Alexandre et al., 2018; Outrequin et al., 2021; n = 16) to obtain a relationship between $^{17}$O-excess$_{phyto}$ and h, assuming that $T_{leaf}$ was 2 °C lower than $T_{air}$:

$$h = 0.25 \, (\pm 0.02) \, ^{17}\text{O-excess}_{phyto} + 130.0 \, (\pm 4.4) \qquad (r^2 = 0.94) \qquad (8)$$

h values reconstructed from $^{17}$O-excess$_{phyto}$ obtained for the three regrowths applying Eq. (8) are in good agreement with corresponding observed daytime averages (Fig. 7b, Table 2). The deviations between reconstructed and measured daytime h values (1 $\pm$ 5 %) are lower than for RH -4 $\pm$ 4 %). However, the difference is insignificant considering the uncertainty on the reconstructed values (4 %). A small amplitude of $\Delta T_{leaf-air}$, as observed in the present study (< 1.1 °C), has thus little impact on the RH estimates from $^{17}$O-excess$_{phyto}$. However, the possibility of larger amplitude, especially in the case of cold forests or warm desert areas, should be considered when interpreting $^{17}$O-excess$_{phyto}$ in terms of RH.

## 4.4 Future tracks for reconstruction of past RH from $^{17}$O-excess of phytoliths extracted from soils

Assessing the relationship between $^{17}$O-excess$_{phyto}$ and RH is crucial for accurate reconstructions using phytolith assemblages extracted from sediments, which are supplied by soil phytoliths from the catchment area. Soil phytoliths likely represent several decades of phytolith production. The limited variation of $^{17}$O-excess in meteoric water (Aron et al., 2021; Surma et al., 2021) and atmospheric water vapor, and its insensitivity of $^{17}$O-excess$_{phyto}$ to temperature make it a powerful indicator of RH. The results of the present study reveal that grass leaf phytoliths record daytime RH under the studied eco-climatic conditions but emphasize that daytime vs nighttime stomatal conductance and $\Delta T_{leaf-air}$ need to be considered when interpreting $^{17}$O-excess$_{phyto}$ in terms of RH. In soils, the accurate interpretation of $^{17}$O-excess$_{phyto}$ is further complicated by the mixture of phytoliths from transpiring (leaves, inflorescences) and non-transpiring plant tissues (stems). As previously reported, grass stem phytoliths contribute to less than 10 % dry weight of the above-ground grass silica content (Webb and Longstaffe, 2002; Ding et al., 2008; Alexandre et al., 2019). A simple calculation shows that this contribution should increase $^{17}$O-excess$_{phyto}$ of grass phytolith assemblages extracted from soils by less than 20 per meg relative to an only grass leaf blade phytolith sample, biasing RH estimates obtained from Eq. (7) by less than 5 % towards higher values.

When tree phytoliths contribute to soil phytolith assemblages, globular granulate phytoliths are abundant (Alexandre et al., 2011, 2018; Aleman et al., 2012). This phytolith type is assumed to form in the non-transpiring secondary xylem of the wood (Collura and Neumann, 2017). However, investigation of phytolith assemblages extracted from soils under different vegetation types, including grass savannas, wooded savannas and enclosed savannas developed under similar RH conditions show the same range of $^{17}$O-excess$_{phyto}$ values in agreement with the $^{17}$O-excess$_{phyto}$ vs RH relationship obtained from the growth chamber calibration (Alexandre et al., 2018). This suggests that the FW of the globular granulate phytoliths can be affected by evaporation and calls for further investigation of its anatomical origin.

## 5 Conclusion

$^{17}O$-excess provides useful insights into evaporation processes at the soil-plant-atmosphere interface as it varies little in rainfall and atmospheric water vapor at the annual scale. In this study, a model-data approach was used to interpret the diurnal and

645 seasonal evolution of the triple oxygen isotope composition of *F. arundinacea* bulk grass leaf water. The C-G steady state model associated with a two-pool mixing equation reliably predicts the triple oxygen isotope composition of grass leaf water during daytime, when all model-relevant parameters are measured. The few model-data discrepancies (up to 4 ‰, 9 ‰, 34 per meg for $\delta^{18}O$, d-excess and $^{17}O$-excess, respectively) are likely related to differences between $T_{plot}$ and actual $T_{leaf}$, variations in the fraction of the unevaporated water pool with changes in transpiration (i.e. Péclet effect), and/or slight

differences between measured RH close to the grass plot and actual RH right around the grass leaves. Deviations of the isotope composition of leaf water from steady state at night are well captured by a non-steady state model. These deviations from steady state can also be identified in the $^{17}O$-excess vs $\delta'^{18}O$ system, whereas this is not the case in the d-excess vs $\delta^{18}O$ system. This example shows that measuring the triple oxygen isotope composition of leaf water contributes to a better understanding of water exchange at the soil-plant-atmosphere interface.

The ability to measure the grass $T_{leaf}$ showed that $\Delta T_{leaf-air}$ is a key determinant of the isotope composition of leaf water. Under the study conditions, it is close to -2 °C at midday, which is in line with the $\Delta T_{leaf-air}$ previously observed on *F. arundinacea* in climate-controlled growth chambers (Alexandre et al., 2019). To gain further insights into this parameter and its variability according to vegetation and climate types, we recommend IR radiometer measurements with spatial coverage as carried out in the present study.

The first continuous record of atmospheric water vapor including $\delta^{17}O$ measurement at a natural site presented here shows that although $\delta^{17}O$, $\delta^{18}O$ and $\delta^2H$ are highly variable at the daily scale, assuming isotope equilibrium between precipitation and atmospheric water vapor is reasonable for these first order parameters at the monthly and annual scales. The second order parameters (d-excess and $^{17}O$-excess) vary little at the daily, monthly and annual scales and are always close to the equilibrium values estimated from precipitation. Further records of the triple oxygen isotope composition of the atmospheric water vapor,

facilitated by the use of laser spectrometers, and precipitation will help to generalize this result.

The measured values of $^{17}O$-excess$_{phyto}$ and daytime RH fit well with the $^{17}O$-excess$_{phyto}$ vs RH equation established from previous growth chamber experiments (Alexandre et al., 2018; Outrequin et al., 2021). However, we emphasize that the magnitude of nighttime stomatal conductance and transpiration needs to be assessed in each study individually to evaluate if RH reconstructed from $^{17}O$-excess$_{phyto}$ reflects daily or daytime conditions. Small $\Delta T_{leaf-air}$ of less than 2 °C as observed in the

670 present study have little impact on the RH estimates from $^{17}O$-excess$_{phyto}$. However, larger $\Delta T_{leaf-air}$ as common in cold forests or warm desert vegetation should be considered when reconstructing RH using $^{17}O$-excess$_{phyto}$ in these contexts. The insights gained from this study provide important tracks for the interpretation of $^{17}O$-excess of phytoliths accumulated in soils and sediments in terms of RH. The study also confirms the consistency of $^{18}\alpha_{phyto-H2O}$ and $\lambda_{phyto-H2O}$ for grasses, which implies that

the distribution of phytoliths along grass leaf blades is virtually invariant. This also opens perspectives for reconstructing past changes in leaf water isotope composition from the triple oxygen isotope composition of fossil grass phytolith assemblages recovered from buried soils and sediments, e.g., useful for land-surface model and data comparisons.

**Appendices**

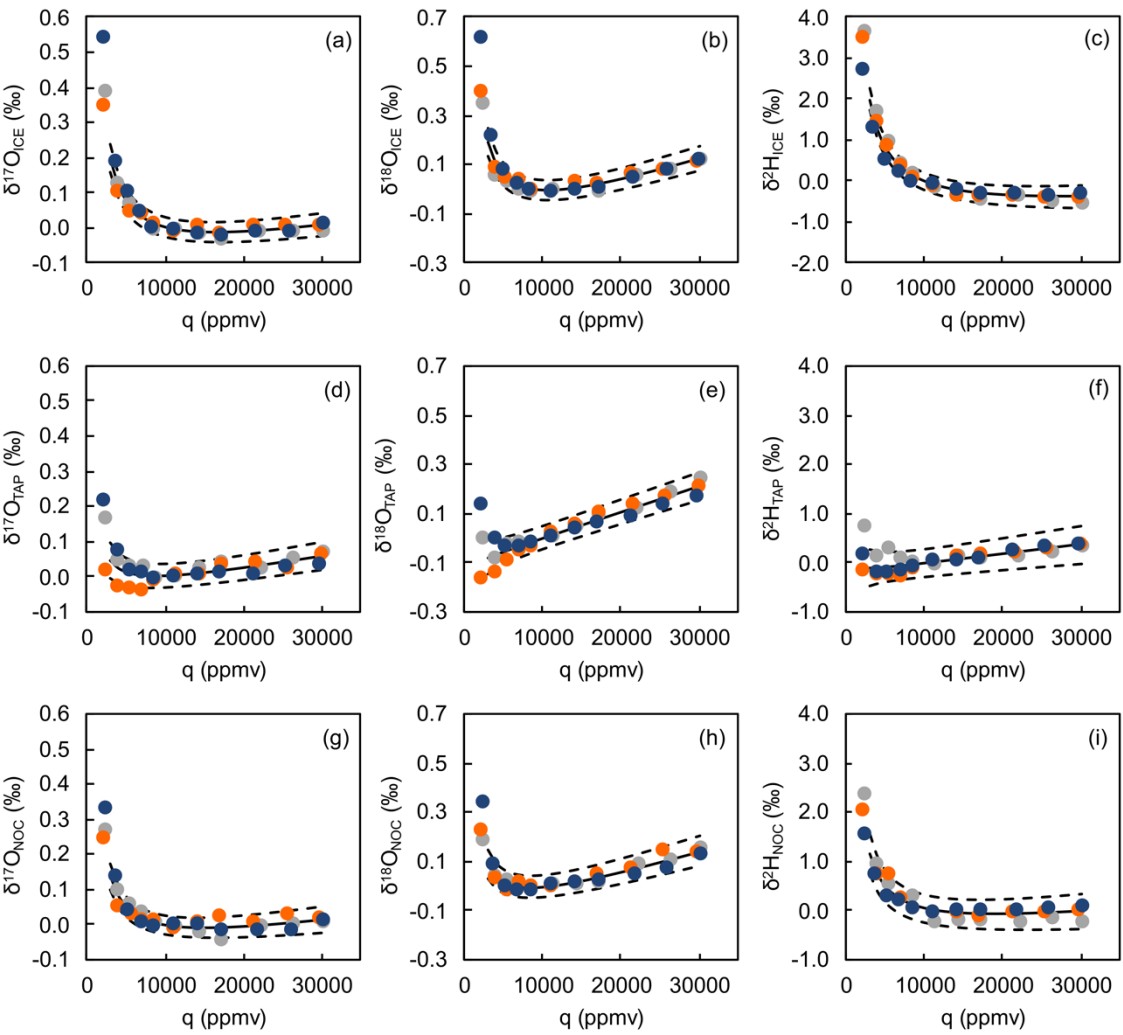

**Figure A1:** Water mixing ratio dependencies of $\delta^{17}O$, $\delta^{18}O$ and $\delta^2H$ normalized to the isotope composition measured at a water mixing ratio (q) of 10000 ppmv for the three water standards ((a)–(c) ICE ($\delta^{18}O$ = -26.85‰), (d)–(f) NOC ($\delta^{18}O$ = -16.91‰), (g)–(i) TAP ($\delta^{18}O$ = -8.64‰)). Mixing ratio dependency calibrations were performed on 26 May 2021 (grey), 20 October 2021 (orange), and 05 January 2022 (blue). Solid and dashed lines show mean and 1 standard deviation of the mixing ratio dependency function.

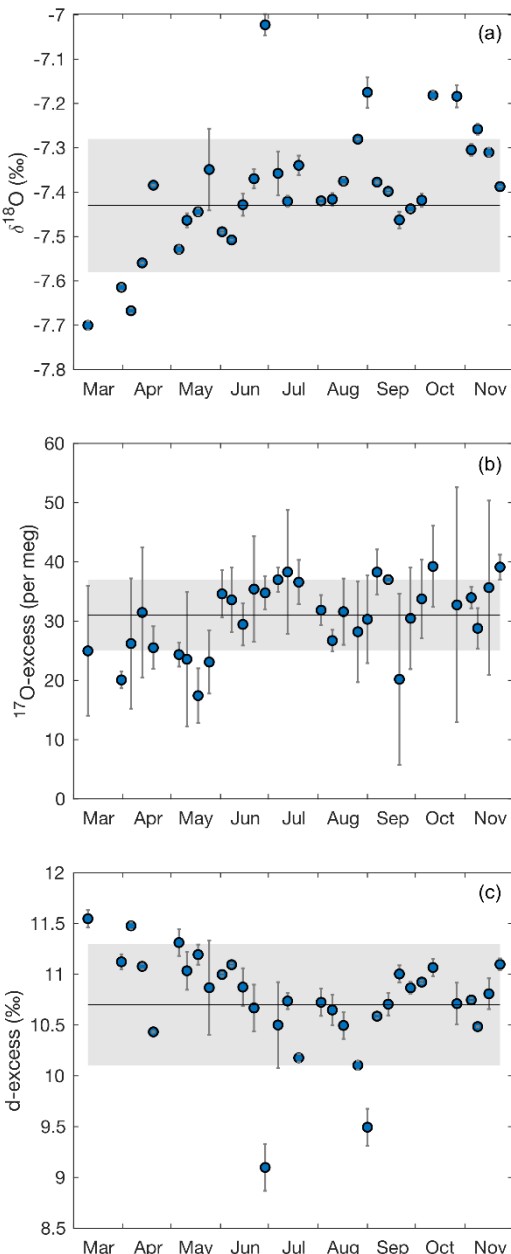

**Fig A2:** Evolution of (a) $\delta^{18}O$, (b) $^{17}O$-excess and (c) d-excess of the irrigation water from March to November 2021. Each data point represents the average isotope composition of the irrigation water over the period between two samples. Error bars are 1 standard deviation (SD). The solid lines and the grey shaded areas indicate mean and SD of the isotope composition of irrigation water averaged over all samples.

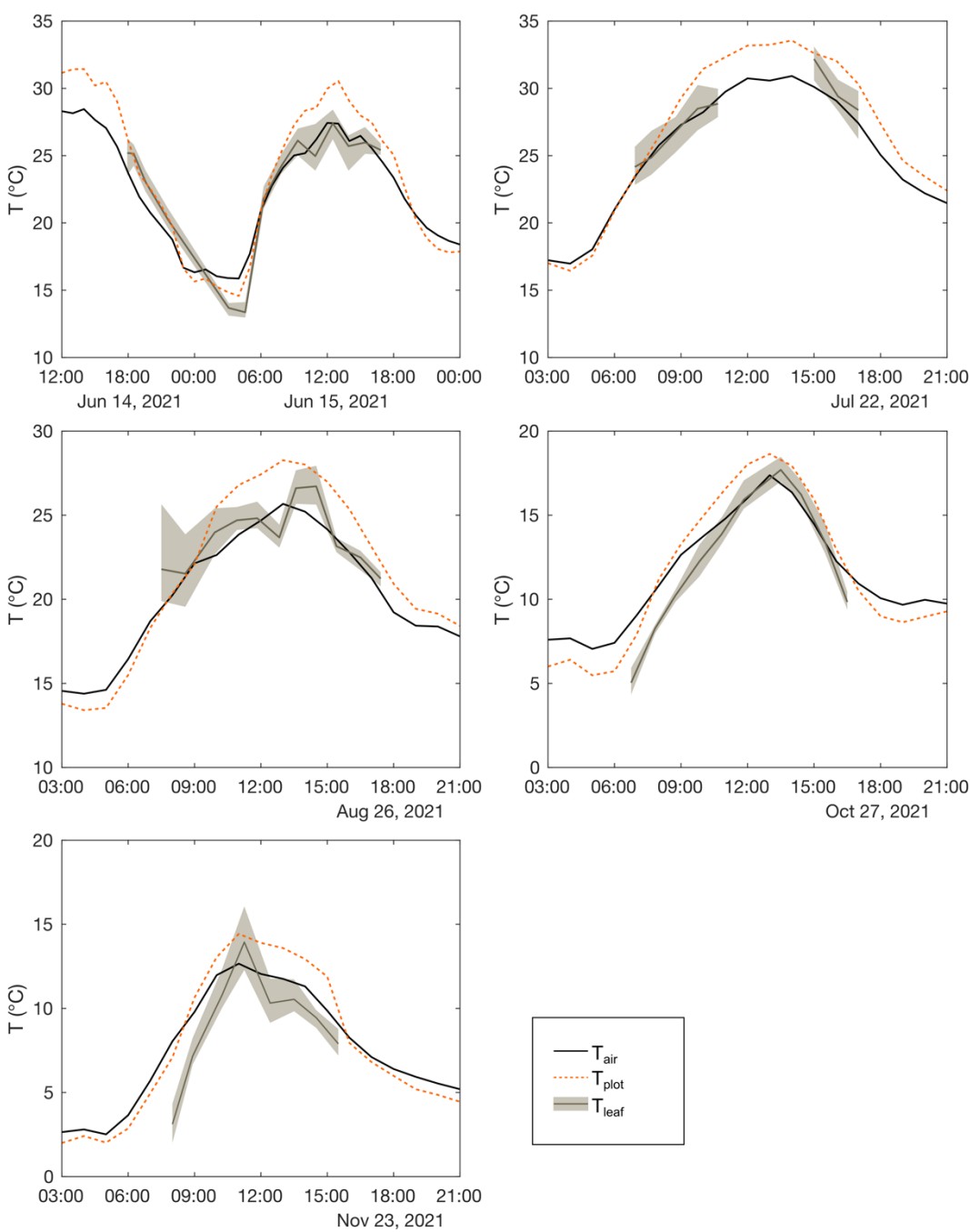

**Figure A3:** Diurnal evolution of atmospheric temperature ($T_{air}$), plot-scale grass leaf temperature ($T_{plot}$) and mean and 1 standard deviation of leaf temperature measurements on single leaves using the Optris IR thermometer ($T_{leaf}$) measured on field days between April and November 2021.

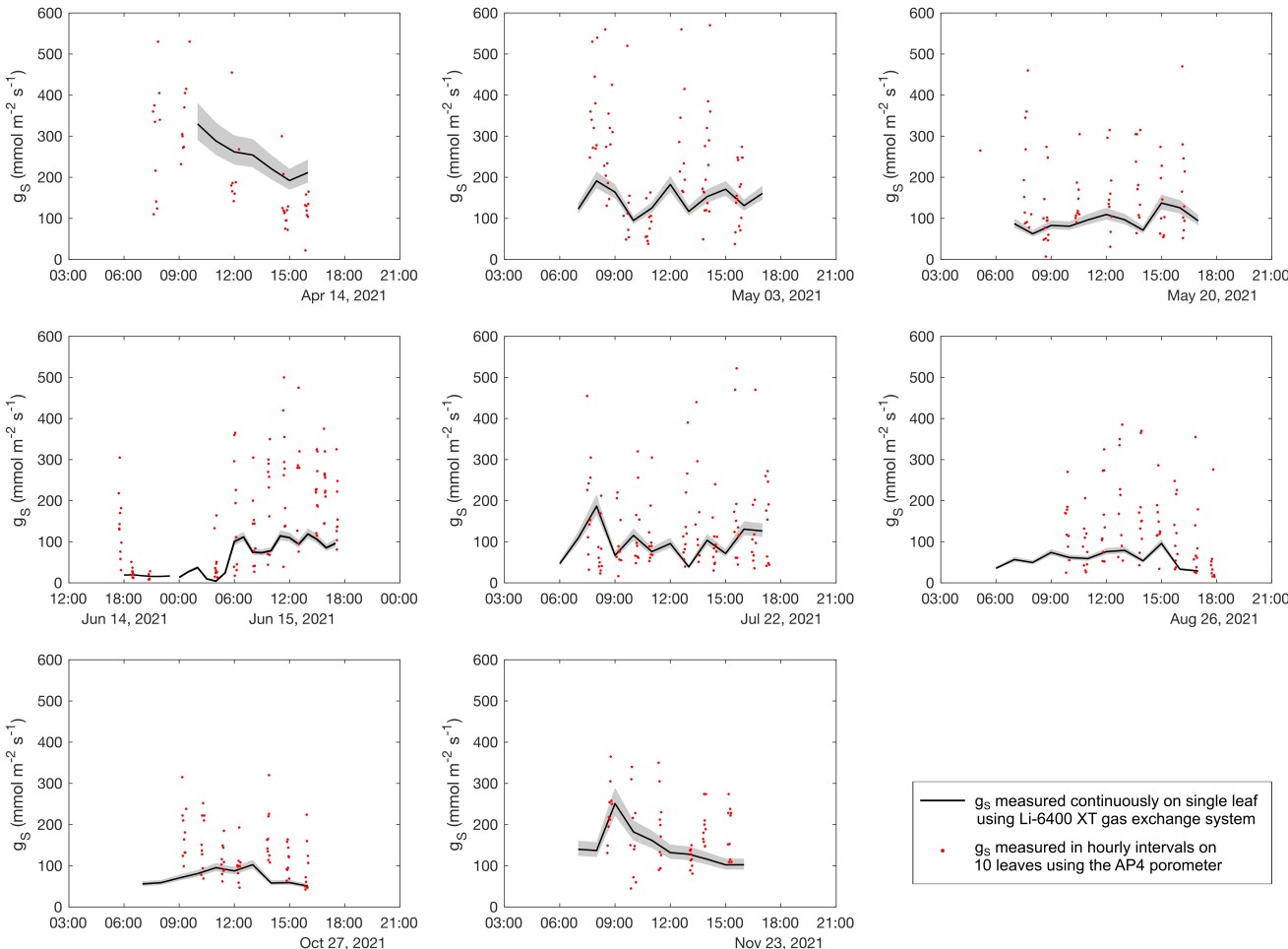

**Figure A4:** Diurnal evolution of stomatal conductance (g$_s$) measured on field days between April and November 2021. Black lines show g$_s$ of a single grass leaf measured continuously over the day using the LI-COR gas exchange system in hourly resolution. Red points represent g$_s$ of different grass leaves measured with the AP4 porometer.

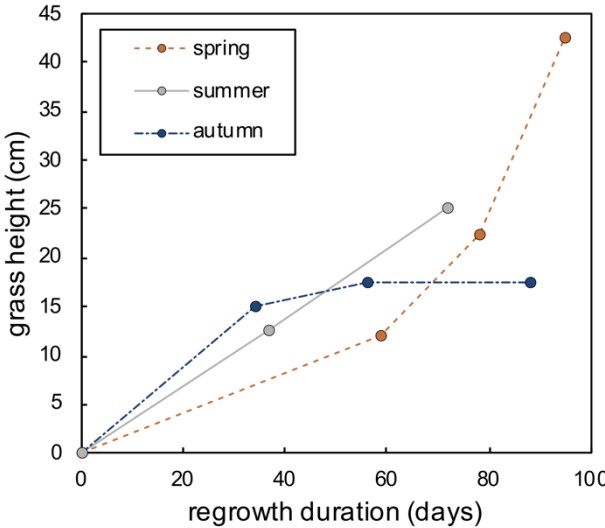

**Figure A5:** Evolution of the grass height over the regrowth duration from 17 February–20 May 2021 (spring), from 15 June–27 August 2021 (summer) and from 27 August–23 November 2021 (autumn).

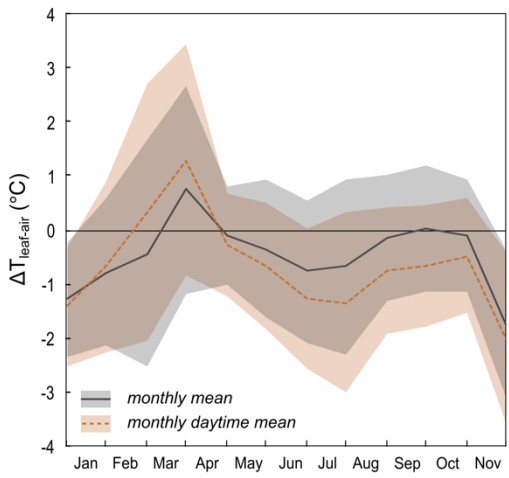

**Figure A6:** Monthly mean and daytime mean of the difference between plot-scale grass leaf temperature ($T_{plot}$) and air temperature ($T_{air}$). The shaded area represents 1 standard deviation.

**Data availability**

Data available within the article or its supplementary materials. Additional data (e.g., LI-COR measurement data) will be made available on request. Meteorological data can be accessed from the COOPERATE database: https://cooperate.eccorev.fr/db.

## Author contribution

AA conceptualized the project and acquired financial support. AA, CV, CVC, IR, JPO and CP designed the experiments and carried out field work. CV, AA, JCM, CVC, CS, HM, JA, and JO performed laboratory analyses. CV and AA prepared the manuscript with contributions from all co-authors.

## Competing interests

The authors declare that they have no conflict of interest.

## Acknowledgements

We acknowledge Martine Couapel for her help with the field experiments. We thank Nicholas Devert and Sabrina Dubois for performing leaf water extractions at ISPA, Bordeaux, France. We thank Michael Staubwasser, Daniel Herwartz and Mohammed El-Shenawy for permission of and support during leaf water triple oxygen and hydrogen isotope analyses in the Stable Isotope Laboratory at the University of Cologne, Germany. This study was conducted in the frame of the HUMI-17 project supported by the ANR (grant nos. ANR-17-CE01-0002-01), CNRS FR3098 ECCOREV, LABEX OT-Med and OSU-Pytheas. It benefited from the CNRS human and technical resources allocated to the Ecotron Research Infrastructure from the state allocation "Investissement d'Avenir" AnaEE France ANR-11-INBS-0001. We also thank the editor, Marco Lehmann and two anonymous reviewers for their useful comments, which contributed to improve the manuscript.

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
