# Peer review of "Examination of the parameters controlling the triple oxygen isotope composition of grass leaf water and phytoliths at a Mediterranean site: A model-data approach"

_Biogeosciences, 2022_

## Author Comment (AC1)

**REPLY TO REVIEWER #1:**

We are grateful for the detailed review and these useful suggestions provided by Reviewer #1. The provided comments will contribute substantially to improving the paper. We will implement most of the suggestions in the revised version of the manuscript. Please, find below in black the comments of the reviewer, in blue our responses to the comments and how these comments will be addressed in the revised manuscript.

The study by Voigt et al. focuses on the main drivers of temporal variations in the triple oxygen isotopic composition of leaf water and phytoliths. The authors performed a grassland plot experiment and measured key physiological, isotopically, and environmental drivers over the seasonal course and over the course of the diel cycle during two days of the experiment. The authors then compared the measured leaf water isotope values with predicted ones derived from Craig-Gordon based steady state and non-steady isotope models, as well as performed a sensitivity analysis to infer the main cause of isotopic variations in leaf water of grasses. As a novel part, they included high-resolution measurements of water vapour isotopes and measured leaf temperature data to improve the models. Besides, the authors found a relationship between daytime RH and 17O-excess of grass phytoliths.

The paper is overall nicely written, and the methods and results are well presented. Yet, after reading through the complete manuscript, I felt that the conclusions of the paper remain vague and that the discussion on the phytolith part, which is highlighted in the title of the paper, falls brief and short, while the leaf water isotope part, which is not highlighted in the title, is well discussed. Given the imbalance, I feel that some parts of the paper should be revised to match the title or that the title itself should be revised.

We thank both reviewers for pointing out that the discussion on the phytolith part falls short in the current version of the manuscript. We agree on this and will extend the discussion on our results in relation to previously published data, better highlight the advantage of $^{17}$O-excess in comparison to the traditional isotope variables ($\delta^{18}$O, $\delta^2$H, d-excess) and providing tracks for the application of $^{17}$O-excess of phytoliths to reconstruct paleo-RH in the revised version. We will additionally revise the title.

**Major**

Discussion 4.3: I think the start of the paragraph reads nice, but then it's not clear how the RH signal in phytoliths 17O-excess in the current study is linked to previous observations and applications. Did the authors expect to find phytoliths' 17O-excess values to be close to daytime than to daily RH values?

In previous experiments carried out in growth chambers environmental parameters were set constant, without considering day-night cycles. Here, under natural conditions, we investigate whether daytime or daily RH should be considered in the equation linking $^{17}$O-excess$_{phyto}$ and RH. This will be further detailed in the revised manuscript.

How well does 17O-excess perform compared to d18O, which was also measured and is well known to carry RH and VPD signals if derived from plant water and material?

This is an important point, and we thank the reviewer for pointing out that this is not clearly explained in the manuscript. $^{17}$O-excess is little variable in atmospheric water vapor and in rainfall at the seasonal or yearly scale, in contrast to $\delta^{18}$O. $^{17}$O-excess of leaf water and phytoliths is

therefore a more direct tracer for changes in RH. We will discuss the advantages of $^{17}$O-excess compared to $\delta^{18}$O in relation to our observations in the revised manuscript.

Would it also make sense to measure d2h and d-excess in phytoliths (maybe not possible)?

Phytoliths are made of amorphous silica ($SiO_2$ ($H_2O)_n$) in which the hydroxyls (OH) and water molecules ($H_2O$) are exchangeable with the surrounding environment. Thus, these molecules are removed by heating (1100°C) under a $N_2$ flow prior to the isotope analysis. There is no more H or $H_2$ to analyze after this dehydration step. This will be detailed in the revised manuscript.

Despite, combination of phytolith with other established proxies, e.g. $\delta^2$H in leaf waxes or $\delta^{18}$O in cellulose, can be part of future work. Combining multiple proxies will allow to achieve more robust estimates of past RH, and open perspectives to obtain a bigger picture of relations between climate and vegetation in the past.

Are there field studies with spatial or temporal resolution on phytoliths 17O-excess values, providing similar results?

The only data from natural sites are from phytoliths extracted from soils sampled along an aridity transect in Central and West Africa (Alexandre et al., 2018). These data better reflect the daily average RH over the growing season rather than RH in the afternoon. However, the scatter is large, the soil phytoliths assemblages represent a mixture of different vegetation sources and we have no information on leaf-to-air temperature gradients that may significantly bias the RH estimate. In the revised manuscript, we will extend the discussion on the implications of our results for the application of this proxy on soil phytoliths.

The discussion should also consider that the authors have only 3 values for phytoliths 17O-excess, which makes it difficult to set up a relationship with RH.

The objective was to verify that the obtained data are consistent with the equation obtained from the growth chamber calibrations, which is the case. More data is needed, in particular from regions with contrasting daytime vs daily RH, and different natural contexts (tropical forest, savannah grassland, steppe, temperate regions, etc.) to generalize our conclusions. We will provide tracks for future studies in the revised version of the manuscript.

The discussion on leaf transpiration and development should be combined with the one in the result part Line 406-412. I would also recommend incorporating the information in Figure 5.

We agree. This section will be implemented in the discussion in revised manuscript.

**Minor:**

Introduction: I am wondering whether it would be worth highlighting that the 17O-excess approach in leaf water is still rather novel compared to d18O and d2H application In my opinion, that is one of the novel parts of the study, but it does not clearly drop out of the introduction.

We agree. In the revised manuscript, we will better highlight the advantages of $^{17}$O-excess in leaf water and phytoliths in comparison to traditional isotopes ($\delta^{18}$O, $\delta^2$H, d-excess).

Line 21 and 23: If "the" is used, its not clear to which specific model (e.g. the CraigGordon Model) it refers to.

We will specify this in the revised version.

Line 24-25. I think these results are not clearly illustrated in the discussion part and figures.

In the revised manuscript, we will extend the discussion on when we expect $^{17}$O-excess of phytoliths to reflect daily vs daytime and provide tracks for the application of this proxy on paleo-records.

Line 27: Yes, but it provides also new knowledge regarding the climate-sensitivity of leaf water stable isotope variations and their models, which I think is not well highlighted so far.

Our study shows that $^{17}$O-excess of atm water vapor and inflow varies little from diurnal to monthly timescale and thus has little influence in $^{17}$O-excess in leaf water. The $^{17}$O-excess in leaf water is mainly driven by RH. In the revised manuscript, we will highlight these advantages of $^{17}$O-excess in comparison to $\delta^{18}$O.

Line 55: Not all abbreviations of the model are explained here, e.g. Rs, aeq, adiff

All used abbreviations will be explained in the revised manuscript.

Line 43: It might be worth adding the multiplication factor that changes "per mil" into "per meg" for 17O-excess.

In the revised manuscript, we will remove the multiplication factor of 1000 from the equation and specify that $^{17}$O-excess is reported in "per meg", which is 0.001 per mil.

Line 62: Written like that it implies that some of the previously cited publications "neglected" the two-pool idea. I think this statement really depends on the plant species, which is quite diverse through all these studies. Maybe rather highlight, that the two-pool idea is important for grass species (but see Liu et al 2017, doi: 10.1111/nph.14549), where parts of the bulk leaf water pool in grasses do not experience evaporation and thus isotopic enrichment. It should also be highlighted that grasses have large isotopic leaf water gradients from the bottom to the top and that "bulk leaf water" is integrating this gradient. Further, the leaf water isotope gradients are integrated into the d18O of plant compounds (but see various papers from Sternberg, Helliker, and Lehmann on plant carbohydrates). In this regard, do we know whether phytolith formation/synthesis is equal along the grass blades? The link between water and phytolith formation during leaf development and growing season could also be an interesting discussion point, particularly if Figure 5 would be considered for the discussion.

Thank you for pointing out this. Silicification and isotope composition of phytoliths along the grass leaf blade has been investigated in Alexandre et al. (2019). Although the $^{17}$O-excess$_{phyto}$ increase with the distance to the leaf base, the $^{17}$O-excess of the bulk phytoliths can be correctly modeled. In the revised manuscript, we will further discuss the differences between phytoliths with an isotope composition that reflects a bulk water content integrating variations in space and time and the leaf water isotope composition which reflects discrete conditions in space and time.

Line 68-70: I strongly assume this statement refers to the "Peclet" effect. I would thus suggest introducing this "term" here so that discussion and introduction are better "linked" to each other.

Yes, we refer here to the "Péclet" effect. We will specify this in the revised manuscript.

Line 70: Rleaf is not defined yet, right?

All used abbreviations will be explained in the revised manuscript.

Line 80: While I agree with the temporal separation, the example is not 100% clear to me. Assuming that sugars are produced under low RH, then they should carry this climatic information in their oxygen isotopic composition. It these sugars are later on used in the night/during rain, their isotopic composition formed under low RH should be transferred (at least partially) to the cellulose which is synthesized under high RH conditions. Maybe the moss example is a bit out of place. There are many studies on grass species and the isotopic composition of plant carbohydrates, which could be highlighted to make this point clearer (but see papers from groups of Schnyder, Helliker, and Lehmann).

We agree that using an example from a grass species may be more appropriate in view of our manuscript objective. We will address this in the revised manuscript.

Line 110: Maybe define "season" here, because it could be a growing season, but I think the authors mean "spring, summer, autumn".

In the revised version, we will clarify that we refer here to different seasons of the year, and not to the growing season.

Line 120: What is the full name of the species? Is it a C3 or C4 plant? Mono or Dicot? This is important information because d18O in leaf water and cellulose and d2H in n-alkanes have been observed to depend on the physiological and biochemical background (but see different papers from Helliker & Ehleringer 2000 and 2002, and Gamarra et al, 2016, PCE).

We used the C3 grass species *Festuca arundinacea*, the same species that has been used for calibration of the $^{17}$O-excess$_{phyto}$ vs RH relationship in growth chamber experiments. As all grasses, it is a monocot species. We will add these details in the revised manuscript.

Line 120: Why was the study performed within a woodland, which I assume is a forest? How do I need to imagine this plot? A grass plot surrounded by trees? I also assume that the grasses were grown on the topsoil in the woodland and that the topsoil was fertilized, is this correct? Does the grass plot include any replicates?

As outlined in line 105, the grass plot was setup in the understory of a natural Mediterranean downy oak forest. The site was chosen, as it provides the necessary facilities for this extensive monitoring study, including meteorological and plant physiological measurements, and the infrastructure to install the Picarro CRDS instrument for on-site atmospheric water vapor measurements. As outlined in line 121-124, the grasses were grown on the shallow topsoil to which we added potting soil. The plot was fertilized to ensure a sufficient amount of nutriments and bio-available silica. No replicates of the grass plot were conducted. We think that this information is sufficiently covered in the current manuscript, so that no changes will be done during revision.

Line 125: The mean isotope value of the tap water could be provided here.

We prefer to show the mean isotope value of tap water in the results section, as it is done in the current version of the manuscript (line 265). The variability of the tap water isotope composition over the experimental period is illustrated in Fig. A2 of the current version.

Line 145: Maybe add the exact period after "over the day".

The period varied a bit from a sampling day to another, mainly from sunrise to sunset, except for the 24h monitoring. The data is illustrated in Fig. A3 and A4.

Line 158: Where the sampled leaf material fully developed and intact?

Yes, we sampled only fully developed, not senescent leaves. We will specify this in the revised version.

Line 163: What does "grass leaves" reflect? Only grass blades? How much material was harvested by end of the season?

Yes, we refer to grass blades. We will specify this in the revised manuscript. The whole grass plot was harvested by the end of each regrowth, resulting in 120 -150 g dry mass.

Table 1 and 2: I assume that providing the raw d17O results of water and phytoliths does not give any additional information and interpretation is only feasible for 17O-excess?

Yes, the $\delta^{17}O$ doesn't provide directly additional information to $\delta^{18}O$ as deviations from GMWL are small. The $\delta^{17}O$ may be back-calculated with high precision from $^{17}O$-excess and $\delta^{18}O$ using the equation given in the introduction section.

Table 2: How many replicates were measured for the phytolith isotopic composition and what does the standard deviation reflect? I assume the plots were not "repeated" and that the SD is "technical replicates". Finally, how exactly was the SiO2 concentration determined? BTW, I think "rate" is the wrong word here because a change over time was not measured, right? Does the SiO2 concentration reflect the amount of phytoliths per gram biomass (i.e, SiO2 = phytoliths)? Do the long and short-cell phytoliths refer to the different "types" as stated in the method (Line 224)?

The SD is determined based on 4 replicates. We will specify this in table caption in the revised version. The silica contents of harvested grass blades were determined by inductively coupled plasma-atomic emission spectroscopy (ICP-AES). The silicification rate is calculated from the measured $SiO_2$ concentration at the end of the regrowth and the length of the regrowth period, assuming a linear production rate (av. rate). Yes, we assume that $SiO_2$ = phytolith, and LC = refers to the proportion of long cell phytoliths on the sum of short and long cell phytoliths in the sample. We will clarify this in the revised manuscript.

Line 207: Plant water refers to leaf water, or did I miss something, e.g. another plant tissue?

Yes, we refer here to leaf water. We will use "leaf water" throughout the revised manuscript.

Line 208: Maybe consider/acknowledge that isotope fractionation can occur during CVD extraction, but it affects d2h more than the d18O (Chen et al, PNAS). To my knowledge, not much is known for d17O. Yet, also amount effects can change the isotopic composition of CVD-extracted water (Diao et al. 2022, HESS). Here I would simply double-check whether the extracted water content was high enough (e.g. difference of weight of exetainers before and after CVD extraction?).

Water yield after leaf water extraction was $103 \pm 5\%$ in average and always higher than 94%. We will add this information in the revised manuscript.

Line 223: The extraction of the phytoliths should be explained in more detail. Its not clear how the phytoliths have been taken or extracted from the leaf material. How much biomass is needed in order to get a sufficient amount of phytoliths? I assume this is time-consuming and difficult work, which then also explains/justifies the low number of replicates in this study.

Phytoliths were extracted following the "wet digestion" protocol in Table 2 of Corbineau et al. (2013). We will specify this in the revised manuscript but will not provide details on the protocol in the manuscript. Instead, we outline the individual steps here: 1) Grass leaves were cut in cm-sized pieces and dried overnight. 2) Dried samples were immersed in 1 N HCl and heated to 80ºC for 2h, subsequently rinsed with deionized water and dried. 3) Concentrated $H_2SO_4$ was added, and the solution was heated for 5h at 80ºC. 4) 30% $H_2O_2$ is slowly added and again heated for 8h. 5) 65% $HNO_3$ and a pinch of $KClO_3$ is added and the solution is heated at 80ºC for 7h. 6) Before being decanted and rinsed with deionized water, the solution remains unheated overnight. 7) The phytoliths are immersed in 0.001 M KOH solution and heated to 130ºC for 10 min. 8) Finally, samples are rinsed with deionized water and dried.

Line 244: Which values were used for gs and gb and were these derived from own measurements or from the literature?

As described in line 146-147, stomatal and boundary layer conductance were measured continuously on one selected leaf using a Li-64000 XT gas exchange system. Values averaged over 30 min before grass leaf sampling are provided in Table S3. In addition, stomatal conductance was measured hourly on ten randomly selected leaves to assess the spatial variability using an AP4 porometer (Line 148-149). The Li-COR measurements are generally within the range of stomatal conductances observed with the porometer (cf. Fig. A4). We will clarify this when describing the model approach.

Line 259: So no leaf water content data is available for this study?

We determined water yield during leaf water extraction, but unfortunately do not have enough data to calculate absolute leaf water content (W m$^{-2}$).

Line 260: Maybe slightly rephrase, and state that a best-fit model was used to set W.

We will rephrase this sentence in the revised manuscript.

Figure 2, Line 322: add "red" to "Dashed circle". Throughout I would avoid using "yellow" in the figures, because its hardly visible.

We will revise the color schemes for all figures to allow readers with color vision deficiencies to correctly interpret our findings.

Line 335-345: How exactly were d-excess and 17O-excess modeled with the Craig-Gordon model? I assume one needs the input of d2h, d17O, and d18O data, then run the model once for each isotope ratio and combine the data to gain excess estimates (e.g d18O and d2h for d-excess, and d17O and d18O for 17O-excess). Please clarify all this in the method part.

The model calculations performed according to Eq. 1-2 for the C-G steady state model and to Eq. 5a,b for the non-steady state model are presented in the supplement tables S3 and S4, respectively. Model calculations are performed for $\delta^2H$, $\delta^{17}O$ and $\delta^{18}O$, and secondary parameters d-excess and $^{17}O$-excess are derived from predicted primary isotope values. All model input variables have been measured. We will clarify this in the modeling section.

Figure 2, 3: What about d2h? The data is shown in the diurnal cycle, but not in the seasonal cycle.

We will add $\delta^2H$ in the figures in the revised manuscript.

358: I would suggest linking the results more clearly to each panel of Figure 4 a – x.

We agree. In the revised version, we will change the order of the panels according to the main text and add the references to each of the panels in the main text.

Figure 4: It appears that a change in temperature is stronger than a change in RH, but this sounds not right to me. From my own experience, the effect of temperature on d18O values in water and organics is rather secondary and induced via the equilibrium fractionation factor, which typically only varies around 2 per mil for d18O between 10 and 30°C or so. Moreover, why have these values been chosen for the sensitivity analysis, e.g. 5% RH and 2°C? Can we really compare them? Does it reflect an x-percent change per mean of each variable?

It is correct that temperature has generally only little effect on $\delta^{18}O$. However, here, we do not change only temperature, but modify the leaf-to-air temperature gradients. This affects the effective relative humidity (h), which is the ratio of atmospheric water vapor pressure over saturation vapor pressure at leaf temperature. 2°C change in $\Delta T_{leaf-air}$ are common in nature and can have a strong effect on leaf water isotope composition, as illustrated in Fig. 4. We will emphasize this in the revised manuscript and justify the used values for RH.

Line 406-412: This is an interpretation of the results and should be moved to discussion 4.3.

We agree. This section will be implemented in the discussion in revised manuscript.

Figure 5, Line 413-421: This part comes a bit out of the blue and it is not yet nicely incorporated in the discussion on the phytoliths. I would also suggest adding more information on the forming water (FW) model and isotope fractionation factor (alpha and lambda values) in the result part. In which space does the alpha value vary and can they be given for each of the examples? If the purpose of figure 5 is to link 17O-excess and d18O of leaf water with those in the "phytoliths forming water", this should be discussed in more detail.

We thank both reviewers for pointing out that figure 5 is not clear and not well implemented in the discussion of the current manuscript version. In the revised version, we will provide more details on the variability of temperature-dependent isotope equilibrium fractionation factors. Further, we will discuss the implications of these results for RH reconstruction from $^{17}O$-excess of fossil phytolith assemblages.

Line 450: So it seems that the Peclet effect could play a role for the leaf water model in the case of high transpiration. Why not provide some values of the Peclet corrected CG-model for the discussion? I assume all the parameters are available to run this model, right?

We will do some model calculations considering Péclet effect and implement the results in the discussion section in the revised manuscript.

Line 489ff: I agree with the paragraph. Maybe it should be more clearly considered that changes in the water vapour isotopic composition are more rapidly affecting the leaf water isotopic composition, within hours (but Lehmann et al. 2020, PCE for grasses), while the isotopic composition of precipitation has to go through the soil before its taken up by the plant and transported to the leaf.

We thank both reviewers for pointing out that the impact of observed isotope variability in the atmospheric water vapor on leaf water is not well discussed in the current version of the manuscript. We will address this point in the revised version.

Figure 6: The irrigation water point could be dropped for decreasing the x-axis scale and reduce the large space on the left side in the figure.

We think that showing the irrigation water is important to get an impression of the evaporative effect of transpiration on the leaf water isotope composition. We will keep this figure as it is in the revised manuscript.

Line 512-516: Please clarify whether the cited papers and equations are derived from grasses or from other species too. Please also clarify how many measurements of phytoliths 17O-excess were taken to generate the linear models, as well as the RH conditions of this study, to have some context. I assume that the RH conditions were similar to those in the current study.

These data are derived from the same grass species *Festuca arundinacea*. The linear model is based on a total of 16 measurements of $^{17}$O-excess of phytoliths at RH of 40,60 and 80%. We will provide these details in the revised manuscript.

Line 528-531: I think the discussion on the nighttime transpiration/stomatal conductance on tropical tree species goes a bit too far, as the current study focuses on grass species.

We agree. We will revise this section in the revised manuscript.

Line 561: I would be surprised if this is really the "first continuous record" given that the laser spectrometer measuring d2h, d18O, d17O are available for some years. How novel is the data? Please clarify this.

The $^{17}$O-excess of water vapor has been monitored in laboratory experiments (Brady and Hodell, 2021; Outrequin et al., 2021). However, here we present the first continuous record of $^{17}$O-excess of atmospheric water vapor in the natural environment. Continuous high-precision measurements of $^{17}$O-excess atm water vapor measurements by CRDS are complex, highly laborious, and cost-intense (cf. Voigt et al., 2021), and rarely been used so far.

572-574: Maybe I missed it, but how was the temperature non-sensitivity of phytoliths 17O-excess determined? Can the authors refer to a table or figure? Maybe move this to discussion point 4.3.

The impact of leaf-to-air temperature gradients on the $^{17}$O-excess of phytoliths is assessed by comparing atmospheric relative humidity (RH) and the effective relative humidity (h), which is the water vapor pressure ratio between the leaf and the atmosphere (Fig. 7). The difference between reconstructed RH and h is lower than the uncertainty on the reconstructed values (>4%). Thus, we conclude that the small leaf-to-air temperature gradients observed in our study (<1.1ºC) do not significantly impact the RH estimates in our case. In the revised manuscript, we will discuss in more detail how leaf-to-air temperature gradients can affect phytolith isotope composition and when we expect this effect to become significant.

576: "RH proxy that is 17O-excess" reads a bit strange

We will rephrase this in the revised manuscript.

576-579: That's a good point, but this has not been discussed yet. See my comments on Figure 5.

See reply to comment on Figure 5.

Discussion 4.1: Is the CG steady-state model also working for d2H of leaf water in the current study? Maybe its worth adding a sentence on that.

The C-G steady state model prediction also agrees within uncertainty with measured $\delta^2$H. We will implement the $\delta^2$H in figure 2 and 3 in the revised manuscript.

Discussion 4.2: The results suggest that there is a difference in the water vapor influence on the temporal changes in d18O vs. 17O-excess, right (Figure 4)? If correct, maybe this is worth briefly discussing.

This comment builds on previous comments regarding the advantage of 17O-excess vs d18O and the link between isotope variability in atm water vapor and leaf water. Our data show that high variability in $\delta^{18}O_{vapor}$ strongly influence $\delta^{18}O_{leaf}$ (hourly timescale, Fig. 3). However, $^{17}$O-excess$_{vapor}$ varies little compared to the large variations in $^{17}$O-excess$_{leaf}$ with changes in RH (from hourly to monthly timescale). Thus, RH changes estimated from $^{17}$O-excess$_{leaf/phyto}$ are more robust than from $\delta^{18}O_{leaf/phyto}$ as $\delta^{18}O_{leaf}$ also depends strongly on source water / atm water vapor isotope composition. In the revised version, we will discuss the advantages of using $^{17}$O-excess$_{phyto}$ instead of $\delta^{18}O_{phyto}$ to estimate paleo-RH.

---

## Author Comment (AC2)

**REPLY TO REVIEWER #2:**

We appreciate the helpful comments and suggestions of Reviewer #2. The provided comments mainly concern aspects, which were also criticised by Reviewer #1. We will follow these suggestions in the revised version of the manuscript as we have realised that this helps to highlight key messages concerning the application of $^{17}$O-excess of phytoliths for paleo-RH reconstruction. Please, find below in black the comments of the reviewer, in blue our responses to the comments and how these comments will be addressed in the revised manuscript.

The manuscript submitted to Biogeosciences presents results of a grass growth study that occurs in a natural setting and compared the results to controlled growth studies that occurred in plant growth chambers. The study monitored many environmental conditions that fluctuated during the day. The triple oxygen isotope compositions and δD were measured from the irrigation water, soil water, water vapor, leaf water, and the siliceous phytoliths (no δD values for the silica). This was compared to modelled water from the Craig and Gordon model modified for plant growth. Overall, this is a very comprehensive dataset that does an excellent job of describing how water vapor triple oxygen and δD compositions change during the day vs. night. A lot of the study focuses on changes in the water vapor without really connecting the impact on the water vapor and leaf water to the grass phytoliths.

Major/minor comments:

Title: The title may want to be edited to better reflect the manuscript which really addresses changes in humidity and stable isotope compositions of leaf water between daytime and nighttime.

This was also suggested by Reviewer #1. We will change the title to better reflect the content of the revised manuscript.

Lines 418-421: Which differences are different by 1.7‰ and 10 per meg? In Fig. 5, the solid red and green lines have different differences even though they both represent a λ of 0.522. Also, how is 'agreement' defined. As written, this seems qualitative as someone could define agreement in a way such that neither λ 522 or 0.524 are agreement with the predicted water from the Craig and Gordon model.

We here refer to the difference between the average isotope composition of leaf water for the regrowth predicted by the C-G steady state model and the formation water reconstructed from phytoliths when using $\lambda_{phyto-H2O}$ of 0.522 and $^{18}\alpha_{phyto-H2O}$ from Dodd and Sharp (2010). The solid red and green lines differ as different colors represent reconstructed FW using different $^{18}\alpha_{phyto-H2O}$. We agree that "agreement" is not an appropriate term here. We will revise figure 5 and the corresponding results section to clarify the key message.

Figure 5: This graph is a little confusing on what is measured vs. modeled. The predicted leaf water is from the C-G model? If so, please add 'gray circle' to the figure to help the reader understand the figure better Is the formation water calculated from the measured grass leaf phytoliths? If so, why are they connected with a line? Passey and Ji (2019) modelled how water would change in different humidity scenarios. Would modelling how the irrigation/precipitation waters change with evaporation in different humidities be more useful than comparing to equilibrium precipitation of silica?

This was also pointed out by Reviewer #1. Precipitation, irrigation and grass leaf phytoliths represent measured data. The leaf water is predicted by the C-G steady state model for average climate and plant physiological conditions over each of the three regrowth periods. The phytolith

formation water (FW) is reconstructed from the measured phytolith isotope composition using different definitions of $^{18}\alpha_{phyto-H2O}$ (different colors), and different $\lambda_{phyto-H2O}$ of 0.522 and 0.524 (dashed vs solid lines). The figure shows the consistency of fractionation coefficients relating the triple oxygen isotope composition of phytoliths and leaf water with previous studies. Closest "agreement" between FW and predicted leaf water is achieved for $\lambda_{phyto-H2O}$ of 0.522 and $^{18}\alpha_{phyto-H2O}$ from Dodd and Sharp (2010). The remaining deviations can be related to slight variations in the distribution of phytoliths and leaf water along the leaf blade, variable mixing proportions of short cell (unevaporated) vs long cell (evaporated) phytoliths. We will discuss this further in the revised manuscript.

Conclusions: Although it may have been missed, there is no conclusion that clearly defines how the phytoliths could be used to predict relative humidity. As the title reflects that phytoliths record daytime humidity, how far off was the humidity as recorded in the phytoliths vs measured? Perhaps adding a term that compares the difference between the $\Delta'^{17}O$ value (or humidity) and the predicted $\Delta'^{17}O$ value (or humidity) would better show to the reader the usefulness of this proxy.

Thank you for pointing this out. In the revised manuscript we will add a discussion section on future perspectives on the use of $^{17}O$-excess of phytoliths for paleo-RH reconstruction.

Overall, the content of this study is of broad importance and fitting for Biogeosciences after minor revisions to better connect the triple oxygen isotope compositions of the phytoliths to relative humidity and the leaf water.

Passey B. H. and Ji H. (2019) Triple oxygen isotope signatures of evaporation in lake waters and carbonates: A case study from the western United States. Earth and Planetary Science Letters. **518**, 1-12.

---

## Author Response (AR1)

Dear Mrs De Jonge

Thank you for inviting us to revise our manuscript. We are grateful to both you and the two reviewers for your detailed and constructive comments and suggestions. We implemented most of the suggestions in the revised version of the manuscript. A detailed point-by-point response to the reviewer's comments is provided below.

In short, we revised the Introduction to better highlight the novelty and potential of the $^{17}$O-excess parameter and point out the objectives of the manuscript. We have implemented modeling of the leaf water isotope composition combining the Craig-Gordon model with the Péclet effect. All the equations used for modelling of the leaf water isotope composition are now presented in the Modelling section. We revised the discussion to better highlight the advantages of the $^{17}$O-excess over $\delta^{18}$O, point out the impact of leaf-to-air temperature gradients and atmospheric water vapor on leaf water. We extended the section on the phytoliths to make key messages clearer. Further, we have added a subsection on future tracks to highlight the interest of $^{17}$O-excess$_{phyto}$ to reconstruct RH as well as the limitations of the proxy. As suggested by both reviewers, we adapted the title to better represent the content of the manuscript. We have revised the colour schemes used in our figures to allow readers with colour vision deficiencies to correctly interpret our findings.

Yours Sincerely

Claudia Voigt (on behalf of all co-authors)

**LINE-TO-LINE COMMENTS TO REVIEWER #1:**

We are grateful for the in-depth review and the useful suggestions provided by Reviewer #1. The provided comments contributed substantially to improve the paper. We implemented most of the suggestions in the revised version of the manuscript. Please, find below in black the comments of the reviewer, in blue our responses to the comments and in bold orange how these comments are addressed in the revised manuscript.

The study by Voigt et al. focuses on the main drivers of temporal variations in the triple oxygen isotopic composition of leaf water and phytoliths. The authors performed a grassland plot experiment and measured key physiological, isotopically, and environmental drivers over the seasonal course and over the course of the diel cycle during two days of the experiment. The authors then compared the measured leaf water isotope values with predicted ones derived from Craig-Gordon based steady state and non-steady isotope models, as well as performed a sensitivity analysis to infer the main cause of isotopic variations in leaf water of grasses. As a novel part, they included high-resolution measurements of water vapour isotopes and measured leaf temperature data to improve the models. Besides, the authors found a relationship between daytime RH and 17O-excess of grass phytoliths.

The paper is overall nicely written, and the methods and results are well presented. Yet, after reading through the complete manuscript, I felt that the conclusions of the paper remain vague and that the discussion on the phytolith part, which is highlighted in the title of the paper, falls brief and short, while the leaf water isotope part, which is not highlighted in the title, is well discussed. Given the imbalance, I feel that some parts of the paper should be revised to match the title or that the title itself should be revised.

We thank both reviewers for pointing out that the discussion on the phytolith part falls short in the current version of the manuscript. We agreed on this and extended the discussion of our results in relation to previously published data (cf answers to major comments below). We adapted the title to better represent the content of the manuscript:

> **"Examination of the parameters controlling the triple oxygen isotope composition of grass leaf water and phytoliths at a Mediterranean site: A model-data approach"**

**Major**

Discussion 4.3: I think the start of the paragraph reads nice, but then it's not clear how the RH signal in phytoliths 17O-excess in the current study is linked to previous observations and applications. Did the authors expect to find phytoliths' 17O-excess values to be close to daytime than to daily RH values?

In the revised manuscript, we emphasize the objective of the study, in relation with the previous findings (New Line 42–45 and 107–111):

> **"Regarding the phytolith isotope signature, previous calibrations have often been performed in controlled environmental conditions, not representative of the diurnal, daily and seasonal climate variations encountered in the natural environment (Alexandre et al., 2018, 2019; Outrequin et al., 2021). Therefore, the question of the time span (seasonal vs annual, diurnal vs daily) integrated in the phytolith isotope composition remains open."**
>
> **[…]**

"**Recent studies in growth chambers and at natural sites demonstrated that unlike the $\delta^{18}O$, the $^{17}O$-excess of phytoliths ($^{17}O$-excess$_{phyto}$), inherited from the $^{17}O$-excess of leaf water, is controlled by RH around the plant, according to a gradient of $4.3 \pm 0.3$ per meg $\%^{-1}$ (Outrequin et al., 2021). This relationship has been found to be independent of grass leaf length and vegetation type (Alexandre et al., 2018, 2019; Outrequin et al., 2021). Further, the $^{17}O$-excess$_{phyto}$ has been shown to be weakly affected by changes in air temperature or atmospheric $CO_2$ levels (Outrequin et al., 2021).**"

As we examined for the first time whether daily or daytime RH is recorded in $^{17}O$-excess$_{phyto}$, a comparison to previous studies is not possible. However, we further explain its reason (Line 596–598):

"**At night, low stomatal conductance and transpiration measured on *F. arundinacea* likely hamper the silicification due to cell water saturation relative to silica formation during daytime transpiration, explaining that daytime RH determines $^{17}O$-excess$_{phyto}$ (Fig. 7).**"

How well does 17O-excess perform compared to d18O, which was also measured and is well known to carry RH and VPD signals if derived from plant water and material?

In the Introduction (New Line 98–105) of the revised paper, we further referred to previous studies linking $\delta^{18}O$ of phytoliths and climate. We show that multiple environmental factors influence $\delta^{18}O$ compared to $^{17}O$-excess, challenging its utility as a proxy for past RH:

"**Phytoliths are micrometric silica particles that form in temperature-dependent isotope equilibrium with water in living plant tissues within a few hours to days (Perry et al., 1987). In grasses, the majority of phytoliths forms in sheaths and leaves, due to concentration of solutes by transpiration (e.g., Webb and Longstaffe, 2000, 2002). Phytolith morphological assemblages recovered from soils and sediments are used to reconstruct vegetation changes and qualitatively inform on climatic conditions at the time of soil formation (Bremond et al., 2005; Aleman et al., 2012; Nogué et al., 2017). Previous studies investigated the potential of $\delta^{18}O$ of phytoliths as a proxy for paleo-temperature (Webb and Longstaffe, 2000, 2002, 2006; Alexandre et al., 2012). However, accurate temperature reconstruction using this proxy requires an independent estimate of the $\delta^{18}O$ of soil water, and an estimate of the effect of RH and transpiration on $\delta^{18}O$ of leaf water.**"

In the Discussion (New Line 562–565), we emphasize that $^{17}O$-excess is little variable in atmospheric water vapor, in contrast to $\delta^{18}O$. $^{17}O$-excess of leaf water and phytoliths is therefore a more direct tracer for changes in RH:

"**Notably, the variations in $^{17}O$-excess of atmospheric water vapor over the 24-hour monitoring are low (45 per meg) compared to its large variability observed in leaf water (120 per meg) (cf. Table 1, S5). In comparison, $\delta^{18}O$ shows much higher variability in atmospheric water vapor (5 ‰) compared to leaf water (8 ‰) (cf. Table 1, S5).**"

Would it also make sense to measure d2h and d-excess in phytoliths (maybe not possible)?

Phytoliths are made of amorphous silica ($SiO_2$ ($H_2O)_n$) in which the hydroxyls (OH) and water molecules ($H_2O$) that exchange with the surrounding environment are removed prior to analysis (already stated in the previous version and now in New Line 245–246):

"**The phytolith samples (1.6 mg) were dehydrated at 1100 °C under a flow of $N_2$ (Chapligin et al., 2010) to prevent the formation of siloxane from silanol groups during dehydroxylation.**"

Are there field studies with spatial or temporal resolution on phytoliths 17O-excess values, providing similar results?

In the Results (New Line 449–452), we added how the obtained $^{17}$O-excess$_{phyto}$ values compare to previous studies:

> **"These isotope values fall within the range of values observed in previous growth chamber calibrations (Alexandre et al., 2018, 2019; Outrequin et al., 2021). The $^{17}$O-excess$_{phyto}$ coincide with the lower range of values reported for phytoliths extracted from soils in Western and Central Africa (Alexandre et al., 2018)."**

In addition, in the discussion, we have added a paragraph highlighting the interest of $^{17}$O-excess$_{phyto}$ to reconstruct RH as well as the limitations of the proxy (New Line 622–641):

> **"4.4 Future tracks for reconstruction of past RH from $^{17}$O-excess of phytoliths extracted from soils**
>
> **Assessing the relationship between $^{17}$O-excess$_{phyto}$ and RH is crucial for accurate reconstructions from phytolith assemblages extracted from sediments, which are supplied by soil and vegetation phytoliths from the catchment area. Soil phytoliths likely represent several decades of phytolith production. The limited variability of $^{17}$O-excess in meteoric water (Aron et al., 2021; Surma et al., 2021) and atmospheric water vapor, and its insensitivity to temperature make the $^{17}$O-excess of grass leaf phytoliths a powerful indicator of RH. The results of the present study reveal that grass leaf phytoliths record daytime RH under the studied eco-climatic conditions but emphasize that nighttime stomatal conductance and $\Delta T_{leaf-air}$ need to be considered when interpreting $^{17}$O-excess$_{phyto}$ in terms of RH. In soils, the accurate interpretation of $^{17}$O-excess$_{phyto}$ is further complicated by the mixture of phytoliths from transpiring (leaves, inflorescences) and non-transpiring plant tissues (stems). As previously reported, grass stem phytoliths contribute to less than 10 % dry weight of the above-ground grass silica content (Webb and Longstaffe, 2002; Ding et al., 2008; Alexandre et al., 2019). A simple calculation shows that this contribution should increase $^{17}$O-excess$_{phyto}$ of grass phytolith assemblages extracted from soils by less than 20 per meg relative to an only grass leaf blade phytolith sample, biasing RH estimates obtained from Eq. (7) by less than 5 %.**
>
> **When tree phytoliths contribute to soil phytolith assemblages, globular granulate phytoliths are abundant (Alexandre et al., 2011, 2018; Aleman et al., 2012). This phytolith type is assumed to form in the non-transpiring secondary xylem of the wood (Collura and Neumann, 2017). However, investigation of phytolith assemblages extracted from soils under different vegetation types, including grass savannas, wooded savannas and enclosed savannas developed under similar RH conditions show the same range of $^{17}$O-excess$_{phyto}$ values in agreement with the $^{17}$O-excess$_{phyto}$ vs RH relationship obtained from the growth chamber calibration (Alexandre et al., 2018). This suggests that the FW of the globular granulate phytoliths can be affected by evaporation and calls for further investigation of its anatomical origin."**

The discussion should also consider that the authors have only 3 values for phytoliths 17O-excess, which makes it difficult to set up a relationship with RH.

The objective was to verify that the obtained data are consistent with the relationship obtained from the growth chamber calibrations, which is the case. More data is needed, in particular from regions with contrasting daytime vs daily RH, and different natural contexts (tropical forest, savannah grassland, steppe, temperate regions, etc.) to generalize our conclusions. We provide tracks for future studies in the revised version of the manuscript (New Line 624–642).

The discussion on leaf transpiration and development should be combined with the one in the result part Line 406-412. I would also recommend incorporating the information in Figure 5.

We implemented this paragraph in the discussion (New Line 567–581):

**"The triple oxygen isotope composition of bulk grass leaf phytoliths is influenced by their distribution along the leaf blade in relation to the leaf water isotope gradient and to silicification patterns (Alexandre et al., 2019; Outrequin et al., 2021). The triple oxygen isotope gradient along grass leaf blades can be predicted by a string-of-lakes model (Alexandre et al., 2019). However, the triple oxygen isotope composition of the bulk grass leaf water is independent of grass leaf length and predictable by the C-G model combined with the mixing equation (Eq. (4)) (Alexandre et al., 2019) or a Péclet effect. The bulk phytolith FW integrates the whole grass elongation period and is thus different from the sampled bulk leaf waters that only represent a snapshot in time. Short cells are among the first cell types to be silicified, sometimes even before the leaf transpires (Motomura et al., 2004; Kumar et al., 2017). The process is metabolically controlled and does not depend on the transpiration rate. Long cell silicification occurs in a second step in relation to transpiration (Motomura et al., 2004; Kumar et al., 2017). Moreover, in grass leaves, the epidermal cells are produced at the base of the leaf and pushed upward during the growth. Hence, epidermal cells along the leaf blade gather phytoliths that were formed at short and long distances relative to the leaf base, i.e. at isotopically low and high evaporative conditions, respectively. The combination of these two processes likely causes the apparent $\lambda_{phyto-H2O}$ being lower than the established $\theta_{SiO2-H2O}$ (=0.524; Sharp et al., 2016) (Outrequin et al., 2021). The consistency of $\lambda_{phyto-H2O}$ equal to $0.522 \pm 0.001$ observed in this study and in previous calibrations (Outrequin et al., 2021), supports that the silicification patterns are systematic and similar for different climate conditions."**

We also revised Figure 5, now better illustrating that calculated FW and predicted leaf water only coincide when $\lambda_{phyto-H2O}$ of 0.522 is used.

[Figure]

**Figure 5: $^{17}O$-excess vs $\delta'^{18}O$ of amount-weighted annual average precipitation, average irrigation water, and the measured isotope composition of phytoliths extracted from *F. arundinacea* grass leaves harvested on 20 May 2021 (spring), 27 August 2021 (summer), and 23 November 2021 (autumn). Also shown are the formation water (FW) predicted using temperature-dependent $^{18}\alpha_{SiO2-H2O}$ from Dodd and Sharp (2010) and $\lambda_{phyto-H2O}$ of 0.522 or 0.524, and the isotope composition of bulk leaf water predicted by the C-G model for steady state conditions combined with the mixing equation (Eq. (4)) using average daytime boundary conditions for the three regrowth periods (Table 2). Error bars represent analytical precisions (see methods section), except for precipitation, for which the amount-weighted standard deviation is indicated.**

**Minor:**

Introduction: I am wondering whether it would be worth highlighting that the 17O-excess approach in leaf water is still rather novel compared to d18O and d2H application In my opinion, that is one of the novel parts of the study, but it does not clearly drop out of the introduction.

We restructured the introduction. We now first introduce previous findings of leaf water isotope studies using traditional isotopes ($\delta^{18}O$, $\delta^{2}H$, d-excess) (New Line 46–81) and then introduce the $^{17}O$-excess parameter (New Line 82–97). This indeed helped us to better highlight the advantages of the $^{17}O$-excess parameter. Similarly, we added a section on previous studies on the $\delta^{18}O$ of phytoliths (New Line 98–105), before introducing the $^{17}O$-excess of phytoliths (New Line 105–111).

Line 21 and 23: If "the" is used, its not clear to which specific model (e.g. the CraigGordon Model) it refers to.

We changed "the" to "a", cf. New Line 27–28:

> **"Deviations from isotope steady state at night are well represented in the triple oxygen isotope system and predictable by a non-steady state model."**

Line 24-25. I think these results are not clearly illustrated in the discussion part and figures.

In the revised manuscript, we extended the discussion on when we expect $^{17}O$-excess of phytoliths to reflect daily vs daytime and provide tracks for the application of this proxy for past RH reconstructions (see responses to major comments).

Line 27: Yes, but it provides also new knowledge regarding the climate-sensitivity of leaf water stable isotope variations and their models, which I think is not well highlighted so far.

We removed this sentence from the abstract in the revised manuscript.

Line 55: Not all abbreviations of the model are explained here, e.g. Rs, aeq, adiff

In the revised manuscript, we moved these equations to the Methods, where we now explain all used abbreviations (New Line 259–274):

> **"According to the C-G isotope steady state model (Craig and Gordon, 1965; Dongmann et al., 1974; Farquhar et al., 2007; Cernusak et al., 2016), the isotope ratio of the evaporated water pool in the leaf ($R_e$) is:**
>
> $$R_e = \alpha_{eq}\alpha_{diff}(1-h)R_S + \alpha_{eq}hR_V, \qquad (2)$$
>
> **where $R_V$ and $R_S$ denote the isotope ratios ($^{2}H/^{1}H$, $^{17}O/^{16}O$ and $^{18}O/^{16}O$) of atmospheric water vapor and source water, respectively. $h$ is the ratio of the actual vapor pressure in the atmosphere to the saturation vapor pressure inside the leaf (i.e. at leaf temperature, $T_{leaf}$). When the leaf-to-air temperature gradient is small, $h$ is equal to RH. The isotope fractionation during water vapor diffusion in air through the leaf stomata and boundary layer ($\alpha_{diff}$) was estimated as:**
>
> $$\alpha_{diff} = \frac{\alpha_{kin}/g_s + \alpha_{kin}^{2/3}g_b}{1/g_s + 1/g_b} \qquad (3)$$

where $g_s$ and $g_b$ (mol m$^{-2}$ s$^{-1}$) denote the stomatal and leaf boundary layer conductances, and $\alpha_{kin}$ denotes the kinetic isotope fractionation during molecular diffusion of water vapor in air. We took $^{18}\alpha_{kin} = 1.028$ and $^{2}\alpha_{kin} = 1.025$ from Merlivat et al. (1978) for $^{18}O/^{16}O$ and $^{2}H/^{1}H$, respectively. Stomatal and boundary layer conductances measured continuously on a single leaf using the Li-COR gas exchange system (see Section 2.1) are used for modeling. For equilibrium isotope fractionation between water and water vapor, temperature-dependent fractionation factors ($\alpha_{eq}$) for $^{18}O/^{16}O$ and $^{2}H/^{1}H$ reported by Majoube et al. (1971) are used herein. The fractionation factors for $^{17}O/^{16}O$ are derived from those of $^{18}O/^{16}O$ according to $^{17}\alpha = {}^{18}\alpha^{\theta}$ using $\theta_{eq} = 0.529$ for liquid-vapor equilibrium (Barkan and Luz, 2005) and $\theta_{kin} = 0.5185$ for the kinetic fractionation during molecular diffusion (Barkan and Luz, 2007).

Line 43: It might be worth adding the multiplication factor that changes "per mil" into "per meg" for 17O-excess.

This is now stated in New Line 84:

"The small variations in $^{17}O$-excess are usually reported in 'per meg', i.e. 0.001 ‰."

Line 62: Written like that it implies that some of the previously cited publications "neglected" the two-pool idea. I think this statement really depends on the plant species, which is quite diverse through all these studies. Maybe rather highlight, that the two-pool idea is important for grass species (but see Liu et al 2017, doi: 10.1111/nph.14549), where parts of the bulk leaf water pool in grasses do not experience evaporation and thus isotopic enrichment. It should also be highlighted that grasses have large isotopic leaf water gradients from the bottom to the top and that "bulk leaf water" is integrating this gradient. Further, the leaf water isotope gradients are integrated into the d18O of plant compounds (but see various papers from Sternberg, Helliker, and Lehmann on plant carbohydrates). In this regard, do we know whether phytolith formation/synthesis is equal along the grass blades? The link between water and phytolith formation during leaf development and growing season could also be an interesting discussion point, particularly if Figure 5 would be considered for the discussion.

We restructured the introduction, by explaining the C-G model and its modifications (Péclet effect, two-pool mixing), and listing the factors that may contribute to model-data leaf water $\delta^{18}O$ discrepancies in previous studies.

New Line 52–63:

"The C-G model is based on the steady-state assumption, i.e. all water that is lost by evaporation is continuously replenished by xylem water. This assumption neglects small diurnal changes in leaf water content that are expected to result in only 3 % error in the predicted leaf water $\delta^{18}O$ enrichment (Farris and Strain, 1978; Farquhar and Cernusak, 2005). The C-G model also assumes isotope steady state, so that the isotope composition of transpired water matches that of source (xylem) water. To take into account the advection of less evaporated stem water to the evaporation site, as well as the diffusion of the evaporating water back to the leaf lamina, a transpiration-dependent correction, called the Péclet effect, can be added to the C-G model (e.g., Buhay et al., 1996; Helliker and Ehleringer, 2000; Roden et al., 2000; Farquhar and Gan, 2003; Farquhar and Cernusak, 2005; Ripullone et al., 2008; Treydte et al., 2014). For grasses, a two-pool model, including a pristine water pool that coincides to the xylem tissues and an evaporated water pool that corresponds to leaf lamina water has been found to best represent bulk leaf water (Liu et al., 2017; Hirl et al., 2019; Barbour et al., 2021). This mixing effect is independent from transpiration, so that a two-endmember mixing equation is combined with the C-G model (Leaney et al., 1985)."

The impact of large isotope gradients along grass leaf blades on the bulk phytoliths is now discussed in section 4.3 (New Line 567–573):

> **"The triple oxygen isotope gradient along grass leaf blades can be predicted by a string-of-lakes model (Alexandre et al., 2019). However, the triple oxygen isotope composition of the bulk grass leaf water is independent of grass leaf length and predictable by the C-G model combined with the mixing equation (Eq. (4)) (Alexandre et al., 2019) or a Péclet effect. The bulk phytolith FW integrates the whole grass elongation period and is thus different from the sampled bulk leaf waters that only represent a snapshot in time."**

Line 68-70: I strongly assume this statement refers to the "Peclet" effect. I would thus suggest introducing this "term" here so that discussion and introduction are better "linked" to each other.

We specified this in the revised manuscript (New Line 56–60):

> **"To take into account the advection of less evaporated stem water to the evaporation site, as well as the diffusion of the evaporating water back to the leaf lamina, a transpiration-dependent correction, called the Péclet effect, has been suggested to be added to the C-G model (e.g., Buhay et al., 1996; Helliker and Ehleringer, 2000; Roden et al., 2000; Farquhar and Gan, 2003; Farquhar and Cernusak, 2005; Ripullone et al., 2008; Treydte et al., 2014)."**

Line 70: $R_{leaf}$ is not defined yet, right?

In the revised manuscript, we removed $R_{leaf}$ here.

Line 80: While I agree with the temporal separation, the example is not 100% clear to me. Assuming that sugars are produced under low RH, then they should carry this climatic information in their oxygen isotopic composition. It these sugars are later on used in the night/during rain, their isotopic composition formed under low RH should be transferred (at least partially) to the cellulose which is synthesized under high RH conditions. Maybe the moss example is a bit out of place. There are many studies on grass species and the isotopic composition of plant carbohydrates, which could be highlighted to make this point clearer (but see papers from groups of Schnyder, Helliker, and Lehmann).

This example is no longer presented in the revised manuscript.

Line 110: Maybe define "season" here, because it could be a growing season, but I think the authors mean "spring, summer, autumn".

We rephrased this part (New Line 117–118):

> **"Grass leaf blades were collected at midday on eight days in different seasons of the year and over a 24-hour period in June for triple oxygen and hydrogen isotope analysis of bulk leaf waters."**

Line 120: What is the full name of the species? Is it a C3 or C4 plant? Mono or Dicot? This is important information because d18O in leaf water and cellulose and d2H in n-alkanes have been observed to depend on the physiological and biochemical background (but see different papers from Helliker & Ehleringer 2000 and 2002, and Gamarra et al, 2016, PCE).

All grasses are monocot. We now specify that the *Festuca arundinacea* is a C3 species (New Line 125–126):

> **"On 14 February 2021, seeds of the C3 grass species *Festuca arundincaea*, also referred to as tall fescue, were sown (8 g m$^{-2}$) on a 5.5 m$^2$ plot in the understory of an oak-dominated forest."**

Line 120: Why was the study performed within a woodland, which I assume is a forest? How do I need to imagine this plot? A grass plot surrounded by trees? I also assume that the grasses were grown on the topsoil in the woodland and that the topsoil was fertilized, is this correct? Does the grass plot include any replicates?

As outlined in New Line 126, the grass plot was setup in the understory of an oak-dominated forest. The site was chosen, as it provides the necessary facilities for this extensive monitoring study, including meteorological and plant physiological measurements, and the infrastructure to install the Picarro CRDS instrument for on-site atmospheric water vapor measurements. As outlined in New Line 128–131, the grasses were grown on the shallow topsoil to which we added potting soil. The plot was fertilized to ensure a sufficient amount of nutriments and bio-available silica. No replicates of the grass plot were conducted. All this information has been given in the Experimental setup section.

Line 125: The mean isotope value of the tap water could be provided here.

We chose to provide the mean isotope value of tap water in the Results section (New Line 302), where it can be compared with the amount-weighted annual averages of precipitation.

Line 145: Maybe add the exact period after "over the day".

The day period varied from a sampling day to another, mainly from sunrise to sunset, except for the 24h monitoring. The data is illustrated in Fig. A3 and A4, where the exact timing can be inferred from.

Line 158: Where the sampled leaf material fully developed and intact?

We now specify that we sampled only fully developed, not senescent leaves (New Line 165–167):

> **"About ten fully developed, not senescent leaf blades from different tillers evenly distributed over the grass plot were immediately transferred to 12 mL Exetainer vials (Labco, High Wycombe, UK), and stored in a fridge until water extraction and isotope analysis."**

Line 163: What does "grass leaves" reflect? Only grass blades? How much material was harvested by end of the season?

We now specify that we sampled only grass leaf blades (New Line 170–171):

> **"At the end of each regrowth, the grass leave blades from the entire plot were harvested, dried at 50 ºC."**

Table 1 and 2: I assume that providing the raw d17O results of water and phytoliths does not give any additional information and interpretation is only feasible for 17O-excess?

Yes, the $\delta^{17}O$ does not provide additional information to the $\delta^{18}O$. The small deviations from the GMWL are expressed in the $^{17}O$-excess.

Table 2: How many replicates were measured for the phytolith isotopic composition and what does the standard deviation reflect? I assume the plots were not "repeated" and that the SD is "technical replicates". Finally, how exactly was the SiO2 concentration determined? BTW, I think "rate" is the wrong word here because a change over time was not measured, right? Does the SiO2 concentration reflect the amount of phytoliths per gram biomass (i.e, SiO2 = phytoliths)? Do the long and short-cell phytoliths refer to the different "types" as stated in the method (Line 224)?

We now specify that the SD is determined based on 4 replicates and clarify that the silicification rate is calculated from the measured $SiO_2$ concentration at the end of the regrowth and the length of the regrowth period, assuming a linear production rate (av. rate), and LC = refers to the proportion of long cell phytoliths on the sum of short and long cell phytoliths in the sample (New Line 186–191):

> **"LC = proportion of long cell phytoliths on the amount of short and long cell phytoliths in the sample. The silicification rate is inferred from the measured $SiO_2$ concentration in grass leaf blades harvested at the end of the regrowth and the length of the regrowth period, assuming a linear production rate (av. rate). […] SD = 1 standard deviation of four replicate measurements on two consecutive days.**

Line 207: Plant water refers to leaf water, or did I miss something, e.g. another plant tissue?

We changed plant water to grass leaf water throughout the revised manuscript.

Line 208: Maybe consider/acknowledge that isotope fractionation can occur during CVD extraction, but it affects d2h more than the d18O (Chen et al, PNAS). To my knowledge, not much is known for d17O. Yet, also amount effects can change the isotopic composition of CVD-extracted water (Diao et al. 2022, HESS). Here I would simply double-check whether the extracted water content was high enough (e.g. difference of weight of exetainers before and after CVD extraction?).

We added the information on water yield and low isotope fractionation potential during water extraction (New Line 223–228):

> **"Water extraction yield was derived by comparing the volume of water collected (in mL) and the difference of sample weights before and after water extraction (with the exetainer and converted in equivalent mL of water). For our sample set, the average water extraction yield was 103 ± 5 % (102 ± 3 % without one outlier) and average extracted volume was 0.5 ± 0.2 mL, with only one extraction volume below 0.3 mL. Thus, methodological uncertainties linked to cryogenic vacuum distillation should be negligible (Diao et al. 2022)."**

Diao, H., Schuler, P., Goldsmith, G. R., Siegwolf, R. T., Saurer, M., and Lehmann, M. M.: Technical note: On uncertainties in plant water isotopic composition following extraction by cryogenic vacuum distillation, Hydrol Earth Syst Sci, 26, 5835–5847, https://doi.org/10.5194/hess-26-5835-2022, 2022.

Line 223: The extraction of the phytoliths should be explained in more detail. Its not clear how the phytoliths have been taken or extracted from the leaf material. How much biomass is needed in order to get a sufficient amount of phytoliths? I assume this is time-consuming and difficult work, which then also explains/justifies the low number of replicates in this study.

We added some details on the phytolith extraction protocol in the revised manuscript (New Line 171–172 and 239–241):

> **"Between 120 and 150 g of dry matter were obtained for phytolith extraction and analysis."**

**[…]**

"**Phytoliths were extracted following the 'wet digestion'-protocol detailed in Table 2 of Corbineau et al. (2013). The protocol involves treatment of the sample with different chemical agents (HCl, H₂SO₄, H2O₂, HNO₃) to remove organic and carbonate compounds.**"

Line 244: Which values were used for gs and gb and were these derived from own measurements or from the literature?

We now specify this in New Line 269–270:

"**Stomatal and boundary layer conductances measured continuously on a single leaf using the Li-COR gas exchange system (see Section 2.1) are used for modeling.**"

Line 259: So no leaf water content data is available for this study?

We determined water yield during leaf water extraction, but unfortunately do not have enough data to calculate absolute leaf water content (W m$^{-2}$).

Line 260: Maybe slightly rephrase, and state that a best-fit model was used to set W.

We rephrased this sentence in the revised manuscript as follows (New Line 294–295):

"**We adjusted $W$ to fit the observed grass leaf water isotope composition. The best fit was found for $W$ of 6 mol m$^{-2}$.**"

Figure 2, Line 322: add "red" to "Dashed circle". Throughout I would avoid using "yellow" in the figures, because its hardly visible.

We revised the color schemes for all figures to allow readers with color vision deficiencies to correctly interpret our findings and modified respectively the figure captions.

Line 335-345: How exactly were d-excess and 17O-excess modeled with the Craig-Gordon model? I assume one needs the input of d2h, d17O, and d18O data, then run the model once for each isotope ratio and combine the data to gain excess estimates (e.g d18O and d2h for d-excess, and d17O and d18O for 17O-excess). Please clarify all this in the method part.

We now specify this in New Line 296–298:

"**Both steady state and non-steady state model calculations were performed for isotope ratios ($^{2}$H/$^{1}$H, $^{17}$O/$^{16}$O and $^{18}$O/$^{16}$O) independently, and the secondary isotope parameters (d-excess and $^{17}$O-excess) were derived from predicted primary isotope values ($\delta^{17}$O, $\delta^{18}$O, $\delta^{2}$H) using the equations given in Section 1.**"

Figure 2, 3: What about d2h? The data is shown in the diurnal cycle, but not in the seasonal cycle.

We added δ²H in the both figures in the revised manuscript.

[Figure]

358: I would suggest linking the results more clearly to each panel of Figure 4 a – x.

In the revised version, we changed the order of the panels according to the main text. Also, references to each of the panels have been added in the main text.

Figure 4: It appears that a change in temperature is stronger than a change in RH, but this sounds not right to me. From my own experience, the effect of temperature on d18O values in water and organics is rather secondary and induced via the equilibrium fractionation factor, which typically only varies around 2 per mil for d18O between 10 and 30°C or so. Moreover, why have these values been chosen for the sensitivity analysis, e.g. 5% RH and 2°C? Can we really compare them? Does it reflect an x-percent change per mean of each variable?

We emphasize now (New Line 521–527) that the influence of variations in $T_{leaf}$ relative to $T_{air}$ on the isotope composition of leaf water is two-fold and how $\Delta T_{leaf\ -air}$ relates to h (RH):

> **"The influence of variations in $T_{leaf}$ relative to $T_{air}$ on the isotope composition of leaf water is two-fold. On the one hand, changes in $T_{leaf}$ slightly modify the magnitude of equilibrium isotope fractionation at the liquid-vapor interface. A few degrees change in $T_{leaf}$ is however of minor importance for the isotope composition of leaf water. In contrast, changes in $\Delta T_{leaf\ -air}$, associated with changes in $T_{leaf}$, modify the water vapor pressure ratio between the leaf and the atmosphere, i.e. h. For example, a decrease in $T_{leaf}$ from 20 to 18 °C at constant $T_{air}$ of 20 °C, modifies h by 5–10 % for RH ranging from 40 to 80 %. As h is the major driver of isotope variability in leaf water, even little variations in $\Delta T_{leaf\ -air}$ can therefore significantly influence the isotope composition of leaf water (Fig. 4i-l)."**

Line 406-412: This is an interpretation of the results and should be moved to discussion 4.3.

As outlined in the responses to the major comments, we implemented this paragraph in Discussion Section 4.3 of the revised manuscript.

Figure 5, Line 413-421: This part comes a bit out of the blue and it is not yet nicely incorporated in the discussion on the phytoliths. I would also suggest adding more information on the forming

water (FW) model and isotope fractionation factor (alpha and lambda values) in the result part. In which space does the alpha value vary and can they be given for each of the examples? If the purpose of figure 5 is to link 17O-excess and d18O of leaf water with those in the "phytoliths forming water", this should be discussed in more detail.

As outlined in the responses to the major comments, we implemented this paragraph in Discussion Section 4.3 of the revised manuscript and modified Figure 5 to clarify its key message.

Line 450: So it seems that the Peclet effect could play a role for the leaf water model in the case of high transpiration. Why not provide some values of the Peclet corrected CG-model for the discussion? I assume all the parameters are available to run this model, right?

We now provide some calculations considering the Péclet effect in this Discussion section (New Line 495–498):

"Considering the Péclet effect (Eq. (5)) instead of a simple mixing significantly reduces model-data discrepancies by 50–80% and leads to deviations between predicted and observed $\delta^{18}O$ and $^{17}O$-excess of grass leaf water in the afternoon on 15 June 2021 that are lower than 1.1 ‰, and 12 per meg, respectively (Table S5). The Péclet effect can thus explain that the observed triple oxygen isotope composition of leaf water varies less than predicted when transpiration is high."

For this, we now also introduce the Péclet model in the methods section (New Line 280–286):

"Instead of a mixing equation, the Péclet effect can be considered to estimate the bulk leaf water isotope composition (Farquhar and Lloyd, 1993; Farquhar et al., 2007; Holloway-Phillips et al., 2016):

$$R_{leaf,ss} = R_s + (R_e - R_s)\frac{1 - e^{-p}}{p} \qquad (5)$$

With p [= EL/CD] the Péclet number, where L is the effective path length, E is the grass leaf transpiration rate, C is the molar density of liquid water (55500 mol m$^{-3}$), and D is the diffusivity of water (2.3 10$^{-9}$ m$^2$ s$^{-1}$ at 25 ºC). One single value of L was applied for the data set and adjusted to fit the observed grass leaf water isotope composition."

Throughout the revised manuscript, we specify when the mixing equation and when the Péclet model was used to calculate the leaf water isotope composition at steady-state.

Line 489ff: I agree with the paragraph. Maybe it should be more clearly considered that changes in the water vapour isotopic composition are more rapidly affecting the leaf water isotopic composition, within hours (but Lehmann et al. 2020, PCE for grasses), while the isotopic composition of precipitation has to go through the soil before its taken up by the plant and transported to the leaf.

We now discuss the impact of changes in the isotope composition of atmospheric water vapor on the leaf water isotope composition.

New Line 540–542:

"The isotope difference between source water and the atmosphere is another key determinant of the leaf water isotope composition. According to the C-G model (Eq. (2)), the influence of atmospheric water vapor relative to source water becomes increasingly important with increasing h (or RH)."

That diurnal variability of atmospheric water vapor can significantly influence the isotope composition of leaf water is shown figure 4 and stated in New Line 558–560:

> **"At the diurnal scale, primary isotope ratios of atmospheric water vapor can vary strongly, often deviating from the monthly equilibrium value. This can lead to significant model-data discrepancies (Fig. 4)."**

Figure 6: The irrigation water point could be dropped for decreasing the x-axis scale and reduce the large space on the left side in the figure.

We chose to keep this figure as is in the revised manuscript, because it highlights the isotope enrichment of the leaf water relative to the irrigation water.

Line 512-516: Please clarify whether the cited papers and equations are derived from grasses or from other species too. Please also clarify how many measurements of phytoliths 17O-excess were taken to generate the linear models, as well as the RH conditions of this study, to have some context. I assume that the RH conditions were similar to those in the current study.

We modified this paragraph as follows:

New Line 582–584:

> **"The relationship between $^{17}O\text{-excess}_{phyto}$ and RH was previously investigated in two growth chamber experiments where _F. arundinacea_ was grown under different conditions of RH (40-60-80 %) and $T_{air}$ (20-24-28 °C) (Alexandre et al., 2018; Outrequin et al., 2021)."**

New Line 586–587:

> **"The two equations obtained from these experiments were statistically similar (Outrequin et al., 2021). Linear regression through both datasets (n = 16) gives […]"**

Line 528-531: I think the discussion on the nighttime transpiration/stomatal conductance on tropical tree species goes a bit too far, as the current study focuses on grass species.

We think that this part is necessary to understand that transpiration/stomatal conductance is species-dependent and this, if daily or daytime RH is recorded by the $^{17}O$-excess of phytoliths needs to be assessed from case to case.

Line 561: I would be surprised if this is really the "first continuous record" given that the laser spectrometer measuring d2h, d18O, d17O are available for some years. How novel is the data? Please clarify this.

The $^{17}O$-excess of water vapor has been monitored in laboratory experiments (Brady and Hodell, 2021; Outrequin et al., 2021). However, here we present the first continuous record of $^{17}O$-excess of atmospheric water vapor in the natural environment. Continuous high-precision measurements of $^{17}O$-excess atm water vapor measurements by CRDS are complex, highly laborious, and cost-intense (cf. Voigt et al., 2021), and rarely been used so far.

572-574: Maybe I missed it, but how was the temperature non-sensitivity of phytoliths 17O-excess determined? Can the authors refer to a table or figure? Maybe move this to discussion point 4.3.

We restructured this section in the revised version to clarify the difference between reconstructed Rh and h. We discuss in more detail the importance of leaf-to-air temperature gradients on Rh reconstruction from 17O-excess of phytoliths (New Line 607–612):

> "RH estimated from $^{17}$O-excess$_{phyto}$ can be biased by variations in $\Delta T_{leaf-air}$. This is because the isotope composition of leaf water is not directly determined by RH, but rather the water vapor pressure ratio between the leaf and the atmosphere, i.e. h (cf. Eq. 2). As discussed in Section 4.2, $\Delta T_{leaf-air}$ of -2 ℃ lead to h that are 5–10% higher than RH. The calibration line obtained from growth chamber experiments is calibrated for $\Delta T_{leaf-air}$ of -2 ℃ (Outrequin et al., 2021). The lower $\Delta T_{leaf-air}$ ranging from -1.1 ℃ to 0.3 ℃ observed in our study can explain the general underestimation of RH reconstructed from the calibration line (cf. Fig. 7a). The effect of $\Delta T_{leaf-air}$ can be removed when considering h instead of RH."

576: "RH proxy that is 17O-excess" reads a bit strange

We rephrased this sentence in the revised manuscript as follows (New Line 671–673):

> "The insights gained from this study provide important tracks for the interpretation of $^{17}$O-excess of phytoliths accumulated in soils and sediments in terms of RH."

576-579: That's a good point, but this has not been discussed yet. See my comments on Figure 5.

See reply to comment on Figure 5.

Discussion 4.1: Is the CG steady-state model also working for d2H of leaf water in the current study? Maybe its worth adding a sentence on that.

The C-G steady state model prediction also agrees within uncertainty with measured $\delta^2$H. We added the $\delta^2$H in figure 2 and 3 in the revised manuscript.

Discussion 4.2: The results suggest that there is a difference in the water vapor influence on the temporal changes in d18O vs. 17O-excess, right (Figure 4)? If correct, maybe this is worth briefly discussing.

The advantage of 17O-excess over d18O is now highlighted (New Line 562–566):

> "Notably, the variations in $^{17}$O-excess of atmospheric water vapor over the 24-hour monitoring are low (45 per meg) compared to its large variability observed in leaf water (120 per meg) (cf. Table 1, S5). In comparison, $\delta^{18}$O shows much higher variability in atmospheric water vapor (5 ‰) compared to leaf water (8 ‰) (cf. Table 1, S5)."

**LINE-TO-LINE COMMENTS TO REVIEWER #2:**

Again, we appreciate the helpful comments and suggestions of Reviewer #2. We follow these suggestions in the revised version of the manuscript as we have realised that this helps to highlight key messages concerning the application of $^{17}$O-excess of phytoliths for paleo-RH reconstruction. Please, find below in black the comments of the reviewer, in blue our responses to the comments and in bold orange how these comments are addressed in the revised manuscript.

The manuscript submitted to Biogeosciences presents results of a grass growth study that occurs in a natural setting and compared the results to controlled growth studies that occurred in plant growth chambers. The study monitored many environmental conditions that fluctuated during the day. The triple oxygen isotope compositions and δD were measured from the irrigation water, soil water, water vapor, leaf water, and the siliceous phytoliths (no δD values for the silica). This was compared to modelled water from the Craig and Gordon model modified for plant growth. Overall, this is a very comprehensive dataset that does an excellent job of describing how water vapor triple oxygen and δD compositions change during the day vs. night. A lot of the study focuses on changes in the water vapor without really connecting the impact on the water vapor and leaf water to the grass phytoliths.

Major/minor comments:

Title: The title may want to be edited to better reflect the manuscript which really addresses changes in humidity and stable isotope compositions of leaf water between daytime and nighttime.

This was also suggested by Reviewer #1. We changed the title to better reflect the content of the revised manuscript:

> **"Examination of the parameters controlling the triple oxygen isotope composition of grass leaf water and phytoliths at a Mediterranean site: A model-data approach"**

Lines 418-421: Which differences are different by 1.7‰ and 10 per meg? In Fig. 5, the solid red and green lines have different differences even though they both represent a λ of 0.522. Also, how is 'agreement' defined. As written, this seems qualitative as someone could define agreement in a way such that neither λ 522 or 0.524 are agreement with the predicted water from the Craig and Gordon model.

Figure 5: This graph is a little confusing on what is measured vs. modeled. The predicted leaf water is from the C-G model? If so, please add 'gray circle' to the figure to help the reader understand the figure better Is the formation water calculated from the measured grass leaf phytoliths? If so, why are they connected with a line?

We have revised figure 5 and its caption to make it clearer.

[Figure]

**"Figure 5: $^{17}$O-excess vs δ'$^{18}$O of amount-weighted annual average precipitation, average irrigation water, and the measured isotope composition of phytoliths extracted from *F. arundinacea* grass leaves harvested on 20 May 2021 (spring), 27 August 2021 (summer), and 23 November 2021 (autumn). Also shown are the formation water (FW) predicted using temperature-dependent $^{18}α_{SiO2-H2O}$ from Dodd and Sharp (2010) and λ$_{phyto-H2O}$ of 0.522 or 0.524, and the isotope composition of bulk leaf water predicted by the C-G model for steady state conditions combined with the mixing equation (Eq. (4)) using average daytime boundary conditions for the three regrowth periods (Table 2). Error bars represent analytical precisions (see methods section), except for precipitation, for which the amount-weighted standard deviation is indicated."**

The figure shows that the formation water reconstructed from the phytoliths only agrees with the predicted leaf water when using a λ$_{phyto-H2O}$ of 0.522, which is consistent with previous studies (Outrequin et al., 2021). This is presented with more clarity in the result section (New Line 454-466):

**"Numerous studies have investigated the temperature-dependent isotope fractionation between amorphous and/or biogenic silica and their formation water ($^{18}α_{phyto-H2O}$) with variable results (e.g., O'Neil and Clayton, 1964; Knauth and Epstein, 1976; Shemesh et al., 1992; Brandriss et al., 1998; Hu and Clayton, 2003; Dodd, 2011, and many more). Here we use temperature-dependent $^{18}α_{phyto-H2O}$ obtained for the diatom-water pair by Dodd and Sharp (2010) (1.0326 at 20ºC). The triple oxygen isotope exponent between silica and water (q$_{phyto-H2O}$) linking $^{17}α_{phyto-H2O}$ to $^{18}α_{phyto-H2O}$ ($^{17}α = {^{18}α^0}$), has been defined as 0.524 ± 0.0002 for the 5–35°C temperature range (Cao and Liu, 2011; Sharp et al., 2016). However, a different value of 0.522 ± 0.001 was consistently obtained for phytoliths, reproducible regardless of bio-climatic constraints (Outrequin et al., 2021). Using this apparent λ$_{phyto-H2O}$, we calculated the triple oxygen isotope compositions of the forming water (FW) in equilibrium with the phytolith samples obtained from the three regrowths (Fig. 5). The reconstructed triple oxygen isotope composition of FW is close to that estimated for daytime average climate conditions of the three regrowths using the C-G model combined with the mixing equation (Eq. (4)) (Fig. 5). The differences are lower than 1.8 ‰ and 33 per meg for δ'$^{18}$O and $^{17}$O-excess, respectively. Using the same $^{18}α_{phyto-H2O}$, but λ$_{phyto-H2O}$ of 0.524, the $^{17}$O-excess of FW is largely underestimated by 35–60 per meg compared to model predictions (Fig. 5)."**

These results are now discussed in more detail in the Discussion Section 4.3 (New Line 567-58q):

**"The triple oxygen isotope composition of bulk grass leaf phytoliths is influenced by their distribution along the leaf blade in relation to the leaf water isotope gradient and to silicification patterns (Alexandre et al., 2019; Outrequin et al., 2021). The triple oxygen isotope gradient along grass leaf blades can be predicted by a string-of-lakes model**

(Alexandre et al., 2019). However, the triple oxygen isotope composition of the bulk grass leaf water is independent of grass leaf length and predictable by the C-G model combined with the mixing equation (Eq. (4)) (Alexandre et al., 2019) or a Péclet effect. The bulk phytolith FW integrates the whole grass elongation period and is thus different from the sampled bulk leaf waters that only represent a snapshot in time. Short cells are among the first cell types to be silicified, sometimes even before the leaf transpires (Motomura et al., 2004; Kumar et al., 2017). The process is metabolically controlled and does not depend on the transpiration rate. Long cell silicification occurs in a second step in relation to transpiration (Motomura et al., 2004; Kumar et al., 2017). Moreover, in grass leaves, the epidermal cells are produced at the base of the leaf and pushed upward during the growth. Hence, epidermal cells along the leaf blade gather phytoliths that were formed at short and long distances relative to the leaf base, i.e. at isotopically low and high evaporative conditions, respectively. The combination of these two processes likely causes the apparent $\lambda_{phyto-H2O}$ being lower than the established $\theta_{SiO2-H2O}$ (=0.524; Sharp et al., 2016) (Outrequin et al., 2021). The consistency of $\lambda_{phyto-H2O}$ equal to $0.522 \pm 0.001$ observed in this study and in previous calibrations (Outrequin et al., 2021), supports that the silicification patterns are systematic and similar for different climate conditions."

Passey and Ji (2019) modelled how water would change in different humidity scenarios. Would modelling how the irrigation/precipitation waters change with evaporation in different humidities be more useful than comparing to equilibrium precipitation of silica?

We are not sure we understand this comment. As specified in the discussion (New Line 625-626), the limited variation of $^{17}$O-excess in meteoric water (Aron et al., 2021; Surma et al., 2021) and atmospheric water vapor, and the insensitivity of $^{17}$O-excess to temperature make the $^{17}$O-excess of phytoliths a powerful indicator of RH.

"The limited variation of $^{17}$O-excess in meteoric water (Aron et al., 2021; Surma et al., 2021) and atmospheric water vapor, and its insensitivity of $^{17}$O-excess$_{phyto}$ to temperature make it a powerful indicator of RH."

Conclusions: Although it may have been missed, there is no conclusion that clearly defines how the phytoliths could be used to predict relative humidity. As the title reflects that phytoliths record daytime humidity, how far off was the humidity as recorded in the phytoliths vs measured? Perhaps adding a term that compares the difference between the $\Delta'^{17}$O value (or humidity) and the predicted $\Delta'^{17}$O value (or humidity) would better show to the reader the usefulness of this proxy.

As has been discussed in section 4.3 (New Line 582-591), previous calibrations set up a reference relationship between $^{17}$O-excess$_{phyto}$ and RH with the precision assessed (Alexandre et al., 2018; Outrequin et al., 2021). Here the small number of phytolith samples analyzed is only used to question whether this relationship applies to daytime or daily RH.

We added a discussion section 4.4 in the revised manuscript, outlining future perspectives for the use of $^{17}$O-excess of phytoliths for past RH reconstruction (New Line 622-641):

"4.4 Future tracks for reconstruction of past RH from $^{17}$O-excess of phytoliths extracted from soils

Assessing the relationship between $^{17}$O-excess$_{phyto}$ and RH is crucial for accurate reconstructions from phytolith assemblages extracted from sediments, which are supplied by soil phytoliths from the catchment area. Soil phytoliths likely represent several decades of phytolith production. The limited variation of $^{17}$O-excess in meteoric water (Aron et al., 2021; Surma et al., 2021) and atmospheric water vapor, and its insensitivity of $^{17}$O-excess$_{phyto}$ to temperature make it a powerful indicator of RH. The results of the present study reveal that grass leaf phytoliths record daytime RH under the studied eco-climatic conditions but emphasize that nighttime stomatal conductance and $\Delta T_{leaf-air}$ need to be

considered when interpreting $^{17}$O-excess$_{phyto}$ in terms of RH. In soils, the accurate interpretation of $^{17}$O-excess$_{phyto}$ is further complicated by the mixture of phytoliths from transpiring (leaves, inflorescences) and non-transpiring plant tissues (stems). As previously reported, grass stem phytoliths contribute to less than 10 % dry weight of the above-ground grass silica content (Webb and Longstaffe, 2002; Ding et al., 2008; Alexandre et al., 2019). A simple calculation shows that this contribution should increase $^{17}$O-excess$_{phyto}$ of grass phytolith assemblages extracted from soils by less than 20 per meg relative to an only grass leaf blade phytolith sample, biasing RH estimates obtained from Eq. (7) by less than 5 %.

When tree phytoliths contribute to soil phytolith assemblages, globular granulate phytoliths are abundant (Alexandre et al., 2011, 2018; Aleman et al., 2012). This phytolith type is assumed to form in the non-transpiring secondary xylem of the wood (Collura and Neumann, 2017). However, investigation of phytolith assemblages extracted from soils under different vegetation types, including grass savannas, wooded savannas and enclosed savannas developed under similar RH conditions show the same range of $^{17}$O-excess$_{phyto}$ values in agreement with the $^{17}$O-excess$_{phyto}$ vs RH relationship obtained from the growth chamber calibration (Alexandre et al., 2018). This suggests that the FW of the globular granulate phytoliths can be affected by evaporation and calls for further investigation of its anatomical origin."

Overall, the content of this study is of broad importance and fitting for Biogeosciences after minor revisions to better connect the triple oxygen isotope compositions of the phytoliths to relative humidity and the leaf water.

Passey B. H. and Ji H. (2019) Triple oxygen isotope signatures of evaporation in lake waters and carbonates: A case study from the western United States. Earth and Planetary Science Letters. **518**, 1-12.

---

## Author Response (AR2)

Dear Mrs De Jonge

We are grateful to both you and the reviewer for accepting our manuscript for publication in *Biogeosciences*. We clarified the two small instances suggested by the reviewer in the final version of the manuscript.

Yours Sincerely

Claudia Voigt (on behalf of all co-authors)